



# Identification of spikes in continuous ground-based in-situ time series of $CO_2$, $CH_4$ and CO: an extended experiment within the European ICOS-Atmosphere Network

Paolo Cristofanelli[1], Cosimo Fratticioli[1], Lynn Hazan[2], Mali Chariot[2], Cedric Couret[3], Orestis Gazetas[4], Dagmar Kubistin[5], Antti Laitinen[6], Ari Leskinen[7,8], Tuomas Laurila[6], Matthias Lindauer[5], Giovanni Manca[9], Michel Ramonet[3], Pamela Trisolino[1], Martin Steinbacher[10]

[1]Institute of Atmospheric Science and Climate - National Research Council of Italy, Bologna I-40129, Italy
[2]Laboratoire des Sciences du Climat et de l'Environnement, Université Paris-Saclay, 91190 Saint-Aubin, France
[3]German Environment Agency UBA, 82475, Germany
[4]Scottish Universities Environmental Research Centre, Glasgow G75 0QF, United Kingdom
[5]Deutscher Wetterdienst, Hohenpeißenberg Meteorological Observatory, 82383 Hohenpeissenberg, Germany
[6]Finnish Meteorological Institute, Helsinki 00560, Finland
[7]Department of Technical Physics, University of Eastern Finland, FI-70210 Kuopio, Finland
[8]Finnish Meteorological Institute, FI-70210 Kuopio, Finland
[9]Joint Research Centre, Ispra I- 21027, Italy
[10]Swiss Federal Laboratories for Materials Science and Technology Empa, Duebendorf CH-8600, Switzerland

*Correspondence to*: P. Cristofanelli (p.cristofanelli@isac.cnr.it)

**Abstract.** The identification of spikes (i.e. short and high variability in the measured signals due to very local emissions occurring in the proximity of a measurement site) is of interest when using continuous measurements of atmospheric greenhouse gases (GHGs) in different applications like the determination of long-term trends and/or spatial gradients, the inversion experiments devoted to the top-down quantification of GHG surface-atmosphere fluxes, the characterization of local emissions or the quality control of GHG measurements. In this work, we analysed the results provided by two automatic spike identification methods (i.e. the standard deviation of the background - SD and the robust extraction of baseline signal - REBS) for a 2-year dataset of 1-minute in-situ observations of $CO_2$, $CH_4$ and CO at ten different atmospheric sites spanning different environmental conditions (remote, continental, urban).

The sensitivity of the spike detection frequency and its impact on the averaged mole fractions on method parameters was investigated. Results for both methods were compared and evaluated against manual identification of the site Principal Investigators (PIs).

The study showed that, for $CO_2$ and $CH_4$, REBS identified a larger number of spikes than SD and it was less "site-sensitive" than SD. This led to a larger impact of REBS on the time-averaged values of the observed mole fractions for $CO_2$ and $CH_4$. Further, it could be shown that it is challenging to identify one common algorithm/configuration for all the considered sites: method-dependent and setting-dependent differences in the spike detection were observed as a function of the sites, case studies and considered atmospheric species. Neither SD nor REBS appeared to provide a perfect identification of the spike events. The REBS tendency to over-detect the spike occurrence shows limitations when adopting REBS as an operational





method to perform automatic spike detection. REBS should be used only for specific sites, mostly affected by frequent very nearby local emissions. SD appeared to be more selective in identifying spike events and the temporal variabilities of $CO_2$, $CH_4$ and CO were more consistent with that of the original datasets. Further activities are needed for better consolidating the fitness for purposes of the two proposed methods and to compare them with other spike detection techniques.

## 1 Introduction

High-precision continuous measurements are needed to monitor the long-term variability of well mixed greenhouse gases (GHGs) in the atmosphere, which are responsible for a large fraction of the anthropogenic forcing to the climate system (IPCC, 2021). These observations are needed to attribute, quantify and reduce uncertainties about the role of the Earth's surface - atmosphere fluxes in determining long-term changes as well as to investigate the complex interactions between GHG fluxes and climate variability (Byrne, 2020). Atmospheric observations can be effectively used in inversion model

systems to obtain optimized information of the spatial and temporal variations of the net atmospheric fluxes over local, regional and global scales (e.g. Christen, 2014; Palmer et al., 2018; Friedlingstein et al., 2022). For such applications, it is pivotal to operate measurement networks with surface footprints representative-enough of the tagged spatial regions. However single sites should be rarely subject to influences of very local emissions representing confounding signals for the correct evaluation of regional and global fluxes. Moreover, the measurement sites must carry out accurate measurements (in

terms of calibration scales) with high-precision and sufficient time resolution.

In the framework of the Integrated Carbon Observation System research infrastructure (ICOS-RI) (Heiskanen et al., 2022), a pan-European monitoring network providing highly compatible, harmonized and high-precision scientific data on the carbon cycle and greenhouse gases was established. ICOS-RI is organized across three domains (atmosphere, ecosystems and oceans): the atmospheric network is providing continuous in-situ data of carbon dioxide ($CO_2$), methane ($CH_4$) and carbon

monoxide (CO) besides other GHGs (i.e. $N_2O$) and atmospheric variables relevant for the investigation of the European carbon cycle. Since the aim of the ICOS-RI atmospheric network is to be compliant with the World Meteorological Organization (WMO) network compatibility goals (WMO, 2020), the measurements are performed following common guidelines and requirements (ICOS RI, 2020). Also the data treatment follows standardized and centralized procedures: all raw data recorded by the measurement sites are delivered in near-real time (i.e. with a 24-hour delay) to the ICOS

Atmospheric Thematic Center (ATC) for the application of automatic quality checks and the averaging to 1-minute and 1-hour mean values (Hazan et al., 2016). Manual revision of the data is performed by the site principal investigator (PI) by using a common quality assurance/quality check application ("ATC QC") running at the ATC server. After this quality control step and a final common review within the ICOS Atmosphere Monitoring Station Assembly, a fully quality controlled Level-2 dataset is released (ICOS RI, 2022). Common instructions about data validation were defined within

ICOS and data have to be invalidated by the site PI only for objective and testified reasons (such an instrument failures or contamination due to maintenance works).





Very local emissions that might occasionally occur in the proximity of the measurement sites and that can produce short but intense variability in the measured GHG signals (i.e. "spikes") can jeopardize the full value of these atmospheric measurements when e.g. using those to constrain the quantification of GHG atmospheric fluxes at regional and global scales.

Here, "local" refers to emissions occurring in a range of a few km from the site which cause positive short-term spikes with maximum duration from minutes to less than a few hours. These signals are not representative of the regional fluxes within the site surface sensitivity area (Storm et al., 2023) because they are superimposed to the GHG variability resulting from the atmospheric background and the regional signal. As an example, the observation records affected by the influence of nearby emission sources should be filtered out for their use in regional or global inversion experiments (e.g. Bergamaschi et al.,

2022). According to the ICOS flagging instructions, these very local contamination events (engine exhaust, local construction, fires, cars, ...) must be considered valid but with a specific descriptive flag ("non background conditions").

In this work, we considered two automatic spike identification methods: the standard deviation of the background (SD) and the robust extraction of baseline signal (REBS) method. They were already tested in a former study (El Yazidi et al., 2018): basing on this former investigation, the SD method was selected to be operationally implemented in the ICOS-Atmosphere

data processing chain to provide an experimental identification of spike occurrence at the measurement sites (see https://icos-atc.lsce.ipsl.fr/P0030.1). The recent expansion of the ICOS atmospheric network represented the opportunity to test the two methods to a wider number of sites and over more extended time frames. The motivation to evaluate the ability of automatic spike identification methods was mainly threefold. Firstly, there is a strong need for some data users to exclude spikes from their analysis as already mentioned above. For example, inversion experiments devoted to the top-down quantification of

GHG emissions need to use de-spiked records of atmospheric observations because very local processes can neither be resolved by the models nor they are appropriately represented in the emission inventories. Secondly, the identification of local spikes can be useful for the analysis and the characterization of specific local emissions. For instance, by analyzing the spike related with emissions from nearby ship transits, Grönholm et al. (2021) used $CO_2$, $CH_4$ and CO data from the ICOS Utö site (Finland) to provide an improved characterization of the emission ratios of ships powered with Liquified Natural

Gas (LNG). Thirdly, a reliable and effective automatic spike methodology represents a powerful tool on the hands of site PIs when performing routine data quality control. Recently, Hoheisel et al. (2023) and Affolter et al. (2021) analyzed the spike occurrences at high mountain sites to infer the impact of local contaminations related with local human activities (tourism, construction works) to long-term atmospheric measurements of $CO_2$ and $CH_4$.

The present study aimed at investigating the sensitivity of the spike identification on continuous (1-minute time resolution)

observations of $CO_2$, $CH_4$ and CO as a function of different settings of the two tested methods as well as at assessing and documenting the impact of these de-spiking procedures to different temporal aggregations (i.e. hourly, monthly and seasonal averages). Finally, we assessed the effectiveness of the two automatic methods in detecting spikes by investigating case studies and comparing the automatic de-spiking results with those provided by manual spike identification. To do so, the $CO_2$, $CH_4$ and CO observations carried out at 10 atmospheric sites representative for different environmental conditions

(remote, continental, urban) were considered.


## 2 Experimental and Methods

### 2.1 Measurements sites

Subsets of data from remote sites (Utö, Jungfraujoch, Zugspitze, Pic du Midi and Monte Cimone), continental sites
(Karlsruhe, Saclay, Ispra), one semi-urban site (Pujo) and one urban site (Paris Jussieu) were used to of assess the
performance of the spike identification methods in different environmental conditions. Since several sites sample from
different heights above ground and, thus, record multiple time series, 19 different sites/heights were investigated for $CO_2$ and
$CH_4$ and 16 for CO (as there were no CO measurements at PUI during the inspected time period).

### 2.1.1 Utö (UTO, Finland)

The UTO site (59.78° N; 21.37° E) is located on the Utö island on the outskirts of the Archipelago Sea in the Baltic Sea
about 80 km southwest from mainland Finland. The island is about 1 $km^2$ in size, mostly rocky and treeless. Typical
vegetation is low and consists of grass and shrubs. The main part of the Baltic Sea is to the south (500 km), the Gulf of
Finland is to the east (400 km). The ICOS measurements are carried from a cell phone mast at 57 m a.g.l. and 65 m a.s.l.,
about 200 m from the coast (Grönholm et al., 2021).

The population on the island is ∼50 people. Local emissions include mainly small boats and a ferry to the mainland, which
may stay overnight in the harbor. AhvenanmaaÅland, an autonomous region of Finland, is to the northwest at a distance of
70 - 120 km. The regional capital city Mariehamn (11'500 inhabitants) is located 90 km from UTO. Turku (186'000
inhabitants) is 90 km far to the NE. Stockholm is 200 km to W-SW.

### 2.1.2 Jungfraujoch (JFJ, Switzerland)

120 This high alpine site (46.55° N; 7.98° E, 3580 m a.s.l.) is situated on a mountain saddle between the two mountains Jungfrau
(4158 m a.s.l) and Mönch (4099 m a.s.l.). The local wind is channeled due to the topography. Surrounding surfaces are
mostly covered by snow or ice apart from some steep slopes of bare rock. No vegetation or soil is present in the vicinity.
The central laboratory for atmospheric observations is located in the uppermost building of the Jungfraujoch facilities, the
so-called Sphinx observatory, which was established in 1937 (Balsiger and Flückiger, 2016). The Sphinx observatory is also
125 accessible for tourists with a public terrace approximately 10 m below the inlet while the upper part is restricted to scientists.
The closest settlements are the tourist villages Wengen (1'200 inhabitants) and Grindelwald (3'800 inhabitants),
approximately 8 km to the NW and 10 km to the NE, respectively. Both are located about 2500 m below Jungfraujoch.
Interlaken (5'700 inhabitants) is located approx. 3 km below Jungfraujoch and 20 km to the north. Thun (42'600 inhabitants)
and Bern (140'000 inhabitants) are located approximately 35 and 60 km to the NW. The Po basin in Northern Italy is located
130 ∼ 150 km to the SE. The impact of regional contributions to the $CO_2$ signal at Jungfraujoch was recently assessed by Pieber
et al. (2022).





### 2.1.3 Zugspitze (ZSF, Germany)

Mt. Zugspitze is the highest mountain of the German Alps. It is located in southern Germany, about 90 km SW of Munich, at the Austrian border near the town of Garmisch-Partenkirchen. The Environmental Research Station "Schneefernerhaus" (47.42° N, 10.79° E, 2666 m a.s.l.), where the ICOS measurements are carried out (Hoheisel et al., 2023), is located on the southern slope of the Mt. Zugspitze. There is no vegetation on the site and the terrain is bare rocks, covered by snow from October to June. The nearest villages and towns are Ehrwald-Lermoos (6 km far, 2'600 inhabitants, 1000 m a.s.l.), Grainau (7 km, 3'500 Inhabitants, 750 m a.s.l.) and Garmisch (12 km, 29'000 inhabitants, 708 m a.s.l.). The nearest largest urban areas are Innsbruck (38 km, 133'000 inhabitants, 574 m a.s.l.) and München (92 km, 1'558'000 inhabitants, 519 m a.s.l.). Please note that for ZSF only data during 2021 were used in this experiment.

### 2.1.4 Monte Cimone (CMN, Italy)

The CMN site (44.19° N; 10.70° E) is located at the top of Mt. Cimone (2165 m a.s.l.), the highest peak of the Northern Apennines and it is characterized by a 360° free horizon. The ICOS site is hosted at the "O. Vittori" Observatory (Cristofanelli et al., 2018) which is located above the timberline: only some patches of grass can be found on the mountaintop, which is mostly rocky and covered with snow for 6-7 months a year. CMN overlooks the highly industrialized Po basin (towards NW–SE) and northern Tuscany (towards S–NW). The most important urban areas are Modena (185'000 inhabitants, 50 km to the N), Bologna (390'000 inhabitants, 60 km to the NE) and Florence (380'000 inhabitants, 55 km to the SE). Local emission sources can be represented by tractors used for transporting items to the mountain top, helicopters and diesel engines used as emergency power supply at the nearby Air Force observatory. During summer, tourists can access the terrace of the "O. Vittori" Observatory (approximately 5 m below the sampling inlet).

### 2.1.5 Pic du Midi (PDM, France)

This high-mountain site (42.56° N, 0.80° E) is located on the NW side of the Pyrenees in SW of France. Due to the high elevation (2877 m a.s.l.), PDM is affected by air masses from the free troposphere from the Atlantic Ocean. As other mountain sites, PDM can be affected by upslope winds and valley wind circulations especially in summer and early autumn, bringing air from the boundary layer of southwest France (covered by intensive croplands and forests, e.g. Fu et al., 2016). In 2015, a field campaign was carried out at PDM to investigate the impact of a small sewage treatment facility to the atmospheric $CH_4$ observations (El Yazidi et al., 2018). The same dataset was considered in this work to assess the efficacy of the automatic spike detection methods (Section 3.4 and 3.6).

### 2.1.6 Ispra (IPR, Italy)

This site (45.81° N; 8.64° E) is located at the South-Eastern border of Lake Maggiore in a semi-rural area at the NW edge of the Po Valley at a distance of 60-100 km from the alpine mountains.





The ICOS site is located within the premises of the Joint Research Centre Ispra (Putaud et al., 2019), at the border of the village of Ispra (5'300 inhabitants). Local emission sources are represented by two cement factories (8.4 km to the NNE and 5.1 km to the SE). Moreover, in the vicinity of the station, agricultural activities taking place during the livestock farming

can impact the $CH_4$ measurements and in case of stagnant weather conditions (with wind speeds < ~1 m/s) can create spikes exceeding a few ppm (Bergamaschi et al., 2022). IPR is equipped with a concrete tall tower (100 m high) with three sampling levels (40 m, 60 m, 100 m a.g.l.).

### 2.1.7 Jussieu Paris (JUS, France)

The JUS site (48.85° N; 2,36° E) is an urban station located in the center of Paris (5[th] district), on a university campus that

houses about 40'000 students and staff. Paris central area extends by 105 km$^2$ and has 2 million inhabitants while Paris agglomeration has 11 million. The air inlet is located on the roof of the main building (30 m a.g.l.) on one side of a small concrete structure present on the roof. The campus roofs are part of an experimental research platform dedicated to the observation of the chemical and dynamic variabilities of the lower atmosphere and therefore house scientific equipment; they are accessible by technical staff. The surroundings include the Seine river which passes to the NE of the campus, and the

Jardin des Plantes, a park and botanical garden with an area of 27 hectares, on the SE. To the W, the Zamansky tower is the tallest building on campus (90 m a.g.l.) but the rest of the campus as well as the rooftops of Paris are at an elevation of 25 – 30 m a.g.l.

### 2.1.8 Karlsruhe Observatory (KIT, Germany)

The KIT site (49.09 N; 8.42 E) is located in a semi-rural region in the Upper Rhine valley at the North Campus of Karlsruhe

for Technology Campus, 12 km north of Karlsruhe city (300'000 inhabitants) and nearby Eggenstein-Leopoldshafen (2 km W, 15'000 inhabitants). Other smaller towns in the surroundings (< 15 km) are Wörth (~17600 inhabitants), Linkenheim-Hohenstetten (12'000 inhabitants), Stutensee (24'000 inhabitants), Bruchsal (44'000 inhabitants) and Braben-Neudorf (11'500 inhabitants). At the measurement site, the Rhine Valley is about 40 km wide and surrounded by 300 - 400 m hills on both sides. The land use in this area is dominated by agricultural fields (ca. 50 %), forests and green areas cover about 11 %,

villages and traffic about 17 %. Local point sources in the near-distance range (< 20 km) are a coal power plant, an oil refinery, two paper mills and one cement factory (E-PRTR, 2017). The site is equipped by a tall tower with four sampling levels (30 m, 60 m, 100 m, 200 m a.g.l.) (Kohler et al., 2018).

### 2.1.9 Puijo (PUI, Finland)

The PUI site (62.91° N; 27.65° E) is located in the city of Kuopio, on top of a 75 m observation and radio transmittance

tower. The tower stands on a hill and its base is 149 m above the surrounding lake level, which is 82 m a.s.l.. Two sampling inlets are available: on top of a 10 m mast on the roof of the tower (84 m a.g.l.) and at 47 m a.g.l.. The measurement site is





located at the southern boreal climatic zone, which is characterized by forests with conifer (mostly pine and spruce) and deciduous (mostly birch) trees, an undulating terrain with rocky soil and moderate height hills, and lots of long lakes in the NW-SE direction.

Nearby sources at PUI are represented by the Puijo tower (restaurant and sewerage ventilation, ~10 m below the top inlet, ~27 m above the lower inlet), a district heating plant 3.5 km to SE (~114 m below the top inlet, ~77 m below the lower inlet), a paper mill 5 km to NE (~164 m and ~127 m below the inlets), a highway in the N-S direction (~230 m and 193 m below the inlets) and a waste disposal site 10.5 km to SW. The Puijo tower is also accessible to tourists and has a viewing platform at ~15 m below the top inlet and ~22 m above the lower inlet.

### 2.1.10 Saclay (SAC, France)

The SAC site (48.72° N; 2.14° E) (Lian et al., 2021) is a semi-urban site. This site is surrounded by agricultural fields (47.4%), forests (25.0%), and urban residential areas (22.2%). The site is located ~20 km SW of the Paris center on the Plateau de Saclay. The closest village and small town are Saint-Aubin (700 inhabitants) and Gif-sur-Yvette (approx. 21'400 inhabitants), located 500 m NW and 1 km to the S of the station, respectively. There is a busy road (N118) located 1 km

from the site. SAC is equipped with a tall tower (100 m high) with three sampling levels (15 m, 50 m and 100m a.g.l.), equipped with meteorological sensors.

### 2.2 Measurements methods

Within the ICOS atmospheric network, sampling equipment setup, sampling procedures, calibration strategy and data processing are executed according to highly standardized and well documented procedures (ICOS RI, 2020). In particular, as

reported by Yver Kwok et al. (2015), the instruments providing $CO_2$, $CH_4$ and CO data must be tested at the ICOS Atmospheric Thematic Center (ATC) Metrological Laboratory (MLab) before their use in the network. The list of accepted analyzers is regularly updated to keep up with new technologies that are continuously tested at the ATC. According to ICOS RI (2020), all sampling heights at tall towers (e.g. PUI, IPR, KIT and SAC) should be sampled sequentially within an hour in order to retrieve hourly vertical gradients. This means that for tall towers, for each single sampling height, less than 60 1-

minute records are available for each hour. The switching sampling strategy is slightly different for each measurement site but since the highest site is considered the most important (for getting regional signals suitable for modeling purposes), priority is usually given to the sampling from the uppermost height. Thus, observations from the highest inlet have the largest data coverage within each single hour. $CO_2$, $CH_4$ and CO are measured at all the considered sites but not at PUI. The considered time series spanned the period 2019 – 2020. Exceptions was ZSF, for which the year 2021 was considered (due to

the fact that ZSF joined ICOS in 2021) and PDM, for which data from the 2015 field campaign (El Yazidi et al., 2018) were analyzed. The specific GHG analyzers used at each considered measurement site are reported in Table 1.





### 2.3 Spike detection methods

The SD and REBS methods were applied to the 1-minute datasets of the sites considered in this experiment. These data are generated in intermediate steps within the data processing for the creation of the Level 2 final dataset (Hazan et al., 2016).

**2.3.1 Standard deviation of the background (SD)**

As reported by El Yazidi et al. (2018), the SD method is designed to select the first available data point, which is assumed not to be a spike ($C_{nospike}$). Then, the next data point ($C_i$) in the time series of 1-minute data is evaluated with respect to $C_{nospike}$: spikes are identified when $C_i$ is higher than a threshold defined as

$$C_{nospike} + \alpha \times \sigma \ + \sqrt{n} \times \sigma \tag{1}$$

If $C_i$ is lower than the threshold from Eq. (1), it is considered as "non-spike" and becomes the new reference value. The method was applied on the 1-minute data with two modes, forwards and backwards: the combination of the detected spikes are kept.

In Eq. (1), $\alpha$ is the parameter to control the selection threshold, $n$ is the number of data between $C_{nospike}$ and $C_i$, and the parameter $\sigma$ is the standard deviation of data falling between the first and the third quartile of the data distribution obtained
by considering a 240-hour time window before $C_{nospike}$. The default values of $\alpha$ (1 for $CO_2$ and $CH_4$ and 3 for CO) were based on the results provided by El Yazidi et al. (2018) and are currently implemented in the operational spike detection chain in use for the whole ICOS-Atmosphere network. The sensitivity analyses presented in this work (Sect. 3.1) considered $\alpha = (0.1, 1, 4)$ for $CO_2$ and $CH_4$ and $\alpha = (0.1, 3, 4)$ for CO. In general, larger $\alpha$ values tend to reduce the sensitivity of the algorithm towards spike detection. As the spike detection method were applied per sampling height, a minimum number of valid 1-
minute data was defined for allowing an effective application. Based on the tests done by El Yazidi et al. (2018), a minimum of 4 full contiguous days (5'760 minutes) should be available for the application of SD for a site with only one sampling height (e.g. CMN, JFJ, ZSF, UTO, JUS). This threshold was decreased to 3'000 minutes for a site with two sampling heights (e.g. PUI), to 2'400 minutes for a site with three sampling heights (e.g. IPR, SAC) and to 1'400 minutes for a site with four sampling heights (e.g. KIT).

**2.3.2 Robust extraction of baseline signal (REBS)**

The REBS method (Ruckstuhl et al., 2001; Ruckstuhl et al., 2012) is a statistical method based on a local linear regression of the time series over a moving time window (characterized by a duration called the "bandwidth"), to account for the slow variability of the baseline signal, with outliers lying above the modeled baseline are iteratively discarded. The REBS code run by ATC is based on the rfbaseline application developed in the IDPmisc package (Locher, 2020) for the R environment.
Because the targeted spikes last a few minutes to a few hours, a bandwidth of 60 min was implemented.

The detection of spikes by REBS is based on the calculation of the following threshold:





$$\hat{g}(t_i) + \beta \times \gamma \qquad\qquad (2)$$

A data point $C_i$, which exceeded this threshold value was considered as a spike. In equation (2), $\beta$ is a tuning parameter (for details, see Ruckstuhl et al., 2012). By default, $\beta$ is set to 3 for $CO_2$ and $CH_4$ and 8 for CO. $\gamma$ is a scale parameter that

represents the standard deviation of data below the baseline curve $\hat{g}(t_i)$ which, for this experiment, was calculated by the R rfbaseline function (Locher, 2020) over the previous 24 hours of data. The sensitivity analyses presented in this work (Sect. 3.1) inspected the impact of changing $\beta$ values with $\beta = (1, 3, 10)$ for $CO_2$ and $CH_4$ and $\beta = (1, 8, 10)$ for CO. Larger $\beta$ values tend to reduce the sensitivity towards spike detection. Similar to the SD approach, a minimum number of valid minute data was set for the method application based on the number of sampling heights at each site. For the application of REBS, one

third of the data recorded within a day (524 minutes) was requested to be available for a site with a single sampling height. This value decreased to 273 minutes for a site with two sampling heights, to 218 minutes for a site with three sampling heights and to 127 minutes for a site with four sampling heights.

## 3 Results

### 3.1 Sensitivity of spike detection to the setting parameters

To provide an overview of the impact of the algorithm settings on the amount of observations identified as "spikes", we calculated the percentage of the data selected on the 1-minute dataset for the different sites, methods and setting parameters (see Figures 1-3). To this aim, the two methods were run with the three different configurations reported in Section 2.3.

### 3.1.1 Carbon dioxide ($CO_2$)

By considering the application of the SD algorithm with the "standard" ATC setting (i.e. $\alpha = 1.0$), the averaged percentage of

1-minute $CO_2$ data considered as spikes was mostly below 1% for all the sites and sampling heights: values exceeding 2% were observed for the highest level of KIT and PUI (Figure 1). By adopting $\alpha = 0.1$, the number of detected spikes increased by factors of 2 to 4 as a function of the sites: the most evident changes were observed for CMN, IPR and JUS. By adopting $\alpha = 4.0$, a significant decrease of the detected spikes occurred in comparison with the standard setting (16/18 time series reported an averaged spike fraction lower than 0.5%). For continental sites, a positive tendency was detected for an increase

of the number of spikes as a function of sampling heights: as an example, at KIT a 0.5% spike occurrence was detected at 30 m, which increased to 2.4% at 200 m for the standard setting. Generally, the highest spike occurrences were detected for the highest sampling height at the continental sites.

By considering the application of the REBS algorithm with the "standard" ATC setting (i.e. $\beta = 3$), the averaged percentage of 1-minute $CO_2$ spikes was above 9% for all the sites and sampling heights (Figure 1). By adopting $\beta = 1$, the number of

detected spikes increased by factors from 3 to 7 as a function of the sites: the most evident changes were observed for CMN and JUS. By adopting $\beta = 10$, a significant decrease of the detected spikes occurred in comparison with the standard setting





with 16/18 time series reporting an averaged spike fraction lower than 0.5%. To evaluate how much the fraction of the detected spikes varied as a function of the different sampling locations, we calculated the ratio between the standard deviation ($\sigma$) to the mean values ($m$) of the averaged spike fraction among the time series. A lower sensitivity to the spike frequency was detected for REBS compared to SD: for the standard settings, the ratio $\sigma/m$ was 0.23 for REBS while it was 0.80 for SD.

### 3.1.2 Methane (CH₄)

By considering the application of the SD algorithm with the "standard" ATC setting (i.e. $\alpha = 1.0$), the averaged percentage of 1-minute $CH_4$ spikes was mostly below 2% for all the sites and sampling heights: a frequency exceeding 3% were only observed at IPR (Figure 2). By adopting $\alpha = 0.1$, the number of detected spikes increased by factors from 2 to 4 as a function of the different sites: the most evident changes were observed for JFJ, JUS, UTO and ZSF. By adopting $\alpha = 4.0$, a significant decrease of the detected spikes occurred in comparison with the standard setting with 13/18 time series reporting an averaged spike fraction lower than 0.5%. Generally, the highest spike occurrences were detected for the lowest sampling heights at the continental sites (as an instance at IPR, for the standard setting, a 5.0 % spike occurrence was detected at 40 m, which decreased to 3.6% at 100 m).

By considering the application of REBS with the "standard" ATC setting (i.e. $\beta = 3.0$), the averaged percentage of 1-minute $CH_4$ spikes ranged from 6.6% at ZSF to 22% at IPR (40 m), see Figure 2. By adopting $\beta = 1$, the number of detected spikes increased by factors from 2 to 9 as a function of the locations and sampling heights: the most evident changes were observed for UTO and ZSF. By adopting $\beta = 10$, a significant decrease of the detected spikes occurred in comparison with the standard setting with 10/18 time series reporting an averaged spike fraction lower than 0.5%. Like for $CO_2$, in respect with SD a lower sensitivity of the spike detection was detected for REBS as a function of the sampling site: for the standard settings, the ratio $\sigma/m$ was 0.32 for REBS with respect to 1.0 for SD.

### 3.1.3 Carbon monoxide (CO)

By considering the application of the SD algorithm with the "standard" ATC setting (i.e. $\alpha = 3.0$), the averaged percentage of 1-minute CO spikes was below 0.5% for all the sites and sampling heights (Figure 3). By adopting $\alpha = 0.1$, the number of detected spikes increased by factors from 3 to 44 as a function of the sites: the most evident changes were observed for the remote sites (CMN, JFJ and UTO) and JUS for which the fraction of detected increased more than 20 times. By adopting $\alpha = 4.0$, all the sites showed a fraction of spike lower than 0.2%. Not clear tendencies were detected for the dependence of the spike detection by the sampling heights.

By considering the application of the REBS algorithm with the "standard" ATC setting (i.e. $\beta = 8$), the averaged percentage of 1-minute CO spikes ranged from 0.1% at CMN to 1.2% at IPR (40 m), see Figure 3. By adopting $\beta = 1$, the fraction of detected spikes largely increased (from 51% at KIT to 71% at IPR and UTO). By adopting $\beta = 10$, the number of detected spikes decreased by ≈50% in respect to the standard setting. For the standard setting, a weak decreasing tendency for the





number of spikes was observed by increasing sampling heights (e.g. at IPR the fraction of detected spikes decreased from
1.2% at 40 m to 0.7% at 100 m). The opposite was observed when $\beta = 1$ was considered with the highest sampling levels of
the continental sites reporting a [+5%;+10%] increase of spike detections in respect to the lowest levels. Despite $CO_2$ and
$CH_4$, the site dependence of the spike fractions was comparable between SD and REBS.

### 3.2 Impact of the spike detections on hourly mean values

Because ICOS atmospheric data are typically provided to external users as hourly mean values, it was important to document
how the application of the two de-spiking methods impacted the dataset of hourly mean values generated by the temporal
aggregation of the 1-minute data. To investigate the potential impact of the de-spiking methods to the 1-hour average values
of $CO_2$, $CH_4$ and CO, we calculated the changes in the percentiles of the original data distribution after the application of the
de-spiking methods with the "standard" settings (Figure 4). For this analysis, only the data record for the highest sampling
level was considered for IPR, KIT, PUI and SAC. Specifically, we calculated the arithmetic differences between the data
distribution percentiles ($5^{th}$, $10^{th}$ $50^{th}$, $90^{th}$ and $95^{th}$) obtained for the de-spiked and the original dataset (i.e. positive values
denoted an increase of the percentile value after the application of de-spiking methods). When the application of SD ($\alpha =
1.0$) is considered for $CO_2$, impacts on the percentile values were observed for IPR, KIT, PUI and SAC (Figure 4). For IPR
(and somewhat PUI), the positive differences for the lowest percentile and the negative differences for the higher percentiles
implied a narrowing of the data distribution after de-spiking. For KIT and SAC, negative differences were observed over
almost all the range of the considered percentiles, thus implying a shift of the data distribution towards lower values. The
same tendencies were observed when REBS ($\beta = 3$) was considered but with a higher impact on the percentile change for
IPR. Similar results were found for $CH_4$: again, significant impacts of the de-spiking were observed for IPR, KIT, PUI and
SAC. However, despite the $CO_2$ case, a narrowing of the data distribution after de-spiking was evident for all these sites (but
KIT). When REBS was used, a wider impact was observed among the stations. Besides IPR, KIT, PUI and SAC, decreases
in the upper percentiles were also observed for the remote sites CMN and PDM. A more limited impact was found for CO:
for both SD ($\alpha = 3.0$) and REBS ($\beta = 8$), a narrowing of the data distribution was observed at IPR, SAC and KIT after de-
spiking application.

### 3.3 Impact of the spike detections on averaged monthly values

An important point was to investigate the effect of the de-spiking methods on the monthly mean values of $CO_2$, $CH_4$ and CO,
which are often used to illustrate/determine the long-term variability and trends of these trace gases. To reach this goal, for
each available time series, the differences of the monthly mean values between the de-spiked and the original dataset were
calculated for each algorithm and for each setting (i.e. varying $\alpha$ and $\beta$ values). When referring to the mole fractions,
hereinafter, we used the terms "impact" or "significant" when the differences between de-spiked and original dataset
exceeded the WMO network compatibility goals (i.e. $\pm 0.1$ ppm for $CO_2$, $\pm 2$ ppb for $CH_4$ and CO; see World
Meteorological Organization (2020).


In general, the impact of de-spiking methods and settings on the monthly mean values depends on the site characteristics (remote vs continental), on the considered trace gas and on the sampling height. The Figures 5 to 7 report the box-plot of the differences ($\Delta CO_2$, $\Delta CH_4$ and $\Delta CO$) between the de-spiked and the original monthly mean values. For $CO_2$, the SD method (with $\alpha = (0.1, 1.0)$) showed impacts only at KIT and PUI. On the other hand, for REBS, impacts were evident for all the

continental sites and for remote sites (in this latter case mostly when $\beta = 1$ was adopted); only for $\beta = 10$ no impacts were diagnosed. For $CH_4$, SD showed impacts only at IPR (for all the adopted settings). Similarly to $CO_2$, REBS showed impacts at all the continental sites when $\beta = (1, 3)$ were considered. For remote sites, an impact was diagnosed only for CMN with $\beta = 1$. For CO, significant deviations in respect to the original dataset were observed only for REBS with $\beta = 1$.

To summarize, the remote sites (CMN, JFJ, ZSF, UTO) were less impacted by the de-spiking and the deviations of the de-

spiked datasets in respect to the original ones were mostly within the WMO network compatibility goals. For the continental sites, larger deviations in respect to the original dataset were generally found after de-spiking with REBS, while significant deviations were observed for SD only for a limited number of sites/sampling heights.

**3.4 Impact of the spike detections on diurnal cycles**

We investigated how much the de-spiking methods influenced the seasonal averaged diurnal cycles of $CO_2$, $CH_4$ and CO at

the test sites. To this aim, for each available time series, we calculated the average diurnal cycles for the original as well as for the de-spiked dataset by using the different settings reported in the Section 2.3 (for tall towers only the highest sampling heights were considered) for the individual seasons. Seasons were defined as December 2019 - February 2020 (DJF), March-May (MAM), June-August (JJA) and September-November (SON).

When looking at the results of de-spiking for $CO_2$ (Figure 8), this analysis revealed that the application of SD had no impact

on the shape of the 24-hour mean cycles: the diurnal peaks detected for the original dataset were maintained for the de-spiked datasets. Only for the remote site ZSF an impact was observed during winter. This was attributed to the wrong identification as a spike of a large $CO_2$ increase related with a regional-scale event, which was observed also at the near mountain site of CMN (but without identification of spikes). For $CO_2$, a larger impact was found for REBS. In particular, when $\beta = 3$ was adopted the diurnal peaks were smoothed at JFJ, IPR, KIT, PUI. Moreover, in some cases (see KIT) also the

overall mean diurnal cycle was modified.

For $CH_4$ (Figure 9), the application of the SD algorithm ($\alpha = 1$) led to a smoothing of the original diurnal peaks at IPR, KIT and SAC, while no impacts were evident at the remote sites. REBS ($\beta = 3$) smoothed the original diurnal peaks at IPR and led to a decrease of the average values over the 24 hours. According to the site PI, the diurnal peaks observed around 8:00 AM and 20:00 PM at IPR, which were significantly smoothed by REBS, were related to very local sources of $CH_4$ due to the

systematic venting of cattle farms located in the proximity of the site. Significant impacts by REBS were also evident at KIT and SAC for $CH_4$. Nevertheless, after the application of REBS ($\beta = 3$), the winter diurnal cycle at KIT appeared more "noisy" in respect to the original dataset suggesting the possibility of spike over-detection (13% of data were identified as





spikes for KIT at 200 m a.g.l.). For CO (Figure 10), significant deviations in respect to the original dataset were only observed when REBS with $\beta = 1$ was considered.

## 3.5 Comparison of SD and REBS spike detections during case studies

In this section, we analyze the ability of the SD and REBS methods in detecting spike events during specific case studies selected by the site PIs at JFJ, UTO, IPR, PUI, SAC and JUS. For each of the considered sites, a list of specific periods (lasting from a few days to a few weeks) affected by the occurrence of spikes were provided by the site PIs for $CO_2$, $CH_4$ and CO. SD and REBS were run for the standard configurations as well as for additional values of $\alpha$ (from 0.1 to 4.0) and $\beta$ (from 1 to 10). Then, the spike identifications were inspected and evaluated by the site PIs who also provided explanations for the possible origin of spikes. For each considered site and case study, a short description of the spike identification results was provided, together with the expert assessment about the performance of the two methods. When possible, we also provided an evaluation about which method was in better agreement judged by the expertise of the stations PIs for these specific case studies. To achieve the best agreement between the expert judgement and the automatic de-spiking, we varied the standard configurations ($\alpha$ and $\beta$ values) and provided the optimal method configuration based on the subjective inspection of the de-spiking method results for each case studies. Here, we provide a representative summary of the results by reporting a subset of the analysed case studies, while the rest of them are presented in the supplementary material (Figures SM1-SM7 and Tables SM1-SM6).

Based on the case study analyses, both SD and REBS tended to overestimate spike occurrences with "standard" settings at JFJ (Table SM1). As an example, here we reported the case study for 19 - 21 November 2020 (Figure 11). In this case, SD appeared to perform better than REBS, overall. For $CO_2$, several data points by high variability were detected as spike by SD in the afternoon of 21 November 2020, with a "false" detection on 19 November 2020 when a $CO_2$ increase due to the vertical transport of PBL air masses affected the measurement site. REBS was able to detect all the "high-variability" data points on 21 November but provided a larger number of "false" detections in the previous days (please note that the adoption of $\beta = 8$, here not shown, would reduce the spike overestimation). For $CH_4$, SD was correct in not detecting spikes, while REBS over detected spikes.

Moving to "continental" sites, by inspecting case studies at IPR, SD appeared to detect less spikes than REBS with "standard" settings for $CO_2$ and $CH_4$; by looking to CO also REBS appeared to provide under detection of spikes. As an example, we reported the time series of trace gases at IPR from 2 to 4 July 2019 (Figure 12). A systematic diurnal variability was evident for $CO_2$ with maxima in the morning and minima during afternoon-evening. Several spikes superimposed to this diurnal variability but they were only marginally detected by SD. For $CH_4$, the most part of spike events were identified by SD but only partially, i.e. SD underestimated the lower part of the events. For CO, basically no spikes were detected by SD, while REBS was able to catch only two major events (events occurring on 3 July were missed). On the other hand, it must be noted that REBS identified as spikes many data points not strictly related with spikes but laying on the "flanks" of large





peaks. A test performed on IPR case studies suggested that decreasing β to 3 would increase the effectiveness of REBS in
detecting CO spikes for the considered events.

The situation appeared to be different for the other continental site SAC. As deduced by the inspection of case studies, SD
appeared to have more skills in detecting spikes than REBS when the standard configuration was used. An evident spike
event for $CO_2$ and $CH_4$ occurred at SAC on 9 - 10 January 2019 (Figure 13). The event was diagnosed by SD (for $CO_2$ the

foot of the event was not detected) and by REBS. However, REBS provided an overdetection of spikes during the considered
period: many data points embedded in $CO_2$ and $CH_4$ peaks were wrongly identified as spikes. As for the JFJ case study,
increasing β to 8 (here not shown) would reduce the spike overestimation for REBS.

**3.6 Comparison between automatic and manual spike detections**

In this Section, for a subset of sites (IPR, PUI, UTO), we presented a comparison between the de-spiking operated by the

methods and that made by the PIs of the sites. The test was carried out for a few months of observations (see Table 2). To
have a sufficient number of spikes available for comparison, the subset of sites was selected to have at least 1% of spike
occurrence with respect to the whole dataset. In this context, the site PIs proceeded in making a manual flagging of the data,
by adopting the same criteria and methodologies used during the routine data quality control. The results from this manual
flagging were compared with the automatic flagging made by SD and by REBS. In this exercise, we also considered the $CH_4$

1-minute data recorded in July - August 2018 at Pic du Midi (PDM) that were analyzed by El-Yazidi (2018). Moreover, a
wider range of the parameters α and β for SD and REBS was considered for this case study. In particular we used α = (0.1,
0.5, 1.0, 1.5, 2.0, 2.5, 3.0, 3.5, 4.0) and β = (1, 2, 3, 4, 5, 6, 7, 8, 9, 10). The methods were run for each of these parameters at
IPR, PUI and UTO.

The reported analysis was based on the calculation of metrics, for SD and REBS separately, that are used in order to evaluate

the effectiveness of dichotomous weather forecasts (Thornes and Stephenson, 2001). Dichotomous forecasts are defined as
forecasts that have two possible outcomes at most (e.g., forecasting the occurrence of fog at a given time and location). As
well as for the case of weather forecasts, the SD and REBS methods provided a dichotomous identification of the spikes
occurrence by flagging each 1-minute data as "spike" or "no-spike". After the automatic and the manual spike flagging, 4
possible outcomes were possible for each 1-minute data:

435        A.  Flagged as spike by the automatic method and by the PI;
    B.  Flagged as spike by the automatic method but not flagged as spike by the PI;
    C.  Not flagged as spike by the automatic methods but flagged as spike by the PI;
    D.  Not flagged as spike neither by the automatic methods, nor by the PI;

Starting from these four possible outcomes, the following metrics were calculated for SD and REBS:

    ● The *hit rate* H:

$$H = p(f|o) = A/(A + C) \tag{3}$$





where $p(f|o)$ is the conditional probability to automatically flag a spike under the condition of having manually flagged a spike.

- The *false alarm rate* F:

$$F = p(f|\underline{o}) = B/(B + D) \tag{4}$$

where $p(f|\underline{o})$ is the conditional probability to automatically flag a spike under the condition of not having manually flagged a spike.

- The Bias ($BIAS$):

$$BIAS = (A + B)/(A + C) \tag{5}$$

which is the ratio between the total number of forecasted spikes and the total number of observed spikes and should be as close as possible to 1.

As being based on the calculation of representative metrics and because the results of automatic spike detections were not shared with the site PIs, this exercise would allow a more objective evaluation of the spike detection methods, also providing
information that could be used to potentially identify "optimal" de-spiking configuration at each site.

In this comparison exercise, we considered the spikes manually detected by PIs to be the "real" (or reference) spikes. This choice is related to the fact that PIs were familiar with the site characteristics and were constantly involved in the analysis and validation of the raw data: thus they represented the most authoritative experts to assess the occurrence of spikes at their own sites. However, it can be argued that also the manual selection performed by the site PIs could be affected by some
degree of "inaccuracy" and cannot be considered as a perfect reference. As an instance, the manual flagging can be less accurate when very frequent spikes affect the time series, making the manual selection of spikes very demanding. Moreover, the manual flagging, as any human data screening, could be affected by a certain degree of arbitrariness. Thus the results of this comparison should not be strictly interpreted as an assessment of the quality of the automatic algorithm performance.

The results of this analysis was reported in Table 2 for the application of the methods to the 1-minute data of $CO_2$, $CH_4$ and
CO. In general, it was difficult to find a case for which an optimal agreement existed between the automatic and the manual spike selection. Compared to SD, REBS detected more spikes which were also identified by the site PIs (see the larger H values) but also more events which were not recognised as spikes by the PI manual flagging (see the larger F values). Only for PDM ($CH_4$), the application of SD led to higher H values. By excluding the PDM case for $CH_4$, REBS had much higher BIAS than SD.

A further analysis was then conducted on "high" spikes in order to evaluate the agreement between automatic and manual de-spiking on a subset of spikes that were expected to have a strong impact on the average values of the time series. In order to perform this analysis, "high" spikes were defined as the 1-minute data points whose distance from a 1-hour rolling mean baseline was higher than 0.5 ppm for $CO_2$ and 2 ppb for $CH_4$ and CO. A sensitivity study was performed by changing the value of the spike selection thresholds but not evident deviations were found compared to using 0.5 ppm and 2 ppb.

In respect to the all-spike analysis, both SD and REBS were more effective in catching events identified by the site PIs (see the higher values for H). Especially for REBS, high H values were obtained indicating that REBS detected a large fraction of





the high spikes manually identified by PIs. However, BIAS strongly increased in respect to the analysis that included also lower amplitude spikes (especially for REBS), indicating a strong tendency in over-estimating the number of events in respect to the PI selection.

By inspecting the variations of H, F and BIAS as a function of α and β, we tried to identify an "optimal" algorithm set-up for each site and chemical species (i.e. the best agreement among manual and automatic detection) and we reported them in Table 2 and 3. For $CO_2$ and $CH_4$, under the "all" data selection, for 4 over 7 cases, the best agreement between the two methods was achieved when the α parameter was lowered to 0.1 - 0.5 for SD and the β parameter was increased to values 5 - 7 for REBS. For REBS, this implied a decrease of F to values comparable with SD and a decrease of H to lower values (thus

implying lower effectiveness of REBS in detecting spikes) but increasing the degree of consistency of the results provided by the two methods.

## 4 Discussion and conclusions

In this exercise, we considered a subset of different atmospheric sites (i.e. remote, continental, urban) for which SD and REBS spike detection methods were applied to 1-minute data of $CO_2$, $CH_4$ and CO. Sensitivity studies were performed in

order to compare the impact of the two methods to the original datasets in terms of spike frequency, hourly and monthly mean values as well as seasonal diurnal cycles. Case studies were considered to test the ability of the automatic methods in identifying spikes. Finally, "blind" tests were executed to objectively compare with a dichotomous analysis the agreement between the spikes manually identified by the site PIs and by the automatic methods.

One main outcome of this study was that REBS identified a larger number of spikes than SD and was less "site-sensitive"

than SD for $CO_2$ and $CH_4$. On average, considering all time series, REBS detected about 10-times more spikes than SD. This led to a larger impact of REBS on the monthly averaged values and on the seasonal averaged diurnal cycle of $CO_2$ and $CH_4$: in respect to the original dataset, the application of REBS (with β = (1, 3)) led to significant impacts (i.e. deviations larger than the WMO network compatibility goal) for all the not-remote sites. On the other hand, SD reported impacts only for selected continental sites. For CO, significant deviations in respect to the original dataset were observed only for REBS with

β = 1. As shown by the analysis of the 1-hour datasets, both SD and REBS were able to narrow the original data distribution or to shift it towards lower values.

The application of the automatic methods to the case studies showed that it was challenging to identify one common algorithm/configuration for all the considered sites. Significant differences in the ability of detecting spikes were observed as a function of the sites, events and considered species. The analyses of selected case study (reported in the supplementary

material) would suggest that REBS performed better than SD at specific sites (i.e. IPR and PUI).

The comparison of SD and REBS with the "blind" manual flagging made by the site PIs at four sites (IPR, PDM, PUI and UTO) showed that both the automatic methods were able to select only a portion of the spike events identified by the site PIs (i.e. hit rate lower than 58%). REBS was better than SD in successfully detecting the spikes selected by the manual





identification (hit rate ranged from 0.17 to 0.58 for REBS and from 0.01 to 0.33 from SD) but with the cost of selecting a
very large fraction of data not recognised as spikes by the site PI (false alarm rate ranged from 0.08 to 0.12 for REBS).
However, this did not necessarily imply less accuracy for REBS: especially in the case of frequent spike occurrence, it
cannot be completely ruled out that spike events could be missed by the PI manual identification. When high spikes were
considered, the hit rate increased both for REBS (ranging from 0.21 to 1.00 as a function of the site) and SD (ranging from
0.02 to 0.75) but with REBS strongly overestimating the number of spikes (BIAS > 1) in respect to the manual identification.
In particular, it should be noted that in the case of low frequency of spike occurrence, REBS appeared to strongly over-detect
the spike occurrence (BIAS higher than 10). Interestingly, based on this comparison, it was pointed out that decreasing the α
parameter to 0.1 - 0.5 for SD and increasing the β parameter to 5 - 7 for REBS led to more consistent spike identification
results between the two methods.

For REBS a further test was carried out by increasing the temporal window over which the baseline ĝ(ti) was calculated to
10 and 30 days. The test was carried out by using β =3 for 8 sites (CMN, JFJ, UTO, IPR, KIT, PUI, SAC, JUS). The
considered case studies revealed that increasing the time window led to a strong overestimation in the number of spikes. As
descriptive examples, the Supplementary Material reports two case studies for SAC. Moreover, REBS (with β =3) was also
run on the time series of the 1-minute $CH_4$ standard deviation values (instead of on the 1-minute mean values). The aim of
this test was to assess the ability of REBS in detecting records characterized by high variability at time scales lower than 1
minute (see Supplementary Material). In respect to the standard application on the 1-minute mean values, the adoption of
this configuration significantly increased the number of the detected spikes by 50%-70% (as a function of the site), implying
a large number of "false" spike detections. On the other hand, some obvious spike events were partially missed for a few
sites (CMN, JFJ, PUI, SAC), thus suggesting that running REBS on standard deviation records was not a suitable strategy to
automatically detect spikes.

To summarize, for the considered measurement sites, neither SD nor REBS appeared to provide a perfect identification of
the spike events but SD is less prone in providing spike over-detection that could introduce inconsistencies in the data
record. It was shown that the REBS tendency to over-detect the spike occurrence could lead to significant biases in the
calculation of the monthly and seasonal mean values in respect to the original data record. This would suggest extreme
caution in adopting REBS as an operational method to perform automatic spike detection at the atmospheric ICOS stations.
REBS should be implemented only for specific sites, mostly affected by more or less frequent very nearby local emissions
(like IPR and PUI), where clear benefits in using REBS were demonstrated. The adoption of common documented
standardised method to detect the occurrence of spikes further increases the already high traceability and the transparency of
the ICOS data production chain.

Our results were only in partial agreement with the outcomes by El Yazidi et al. (2018) who applied SD and REBS to a $CH_4$
record which was occasionally subject to very local pollution: this former study reported that REBS had a notable tendency
in not catching a certain fraction of the spikes, while SD correctly detected most of the contaminated data. This suggested
that the effectiveness of the automatic spike detection by SD and REBS could be site specific.



For this reason, further activities are needed for better consolidating the fitness for purposes of the two proposed methods. More information could be obtained by the operational implementation of the automatic de-spiking methods over the whole
ICOS network. This information can be used to adopt future refinements in the method settings devoted to the optimization of the automatic de-spiking. These refinements can include the possibility to combine the $CO_2$, $CH_4$ and CO spike detection and/or to include other diagnostic parameters (e.g. wind speed and direction) and/or to tune/select the methodology as a function of the different sites by adopting approaches like that proposed in Section 3.6. A further action to be pursued is the exchange of experiences with other initiatives or measurement networks in the atmospheric composition landscape (e.g. the
"Aerosol, Clouds and Trace Gases Research Infrastructure" - ACTRIS RI or the "Advanced Global Atmospheric Gases Experiment" - AGAGE) with the aim of considering different or novel (e.g. machine-learning based) spike detection methods.

*Author contributions.* PC, CF, MR, PT designed the study. PC evaluated the data and wrote the paper with the help of CF.
CF, OG, LH and PT made the data analyses. LH set and run the codes of the spike algorithms. MC, CC., DK, AL, AL, TL, GM, MR and MS contributed in the design of the experiment, in the interpretation of analysis results and in the manuscript review/editing. MC, CC, DK, MR, AL, AL, ML, TL, GM, PT and MS were responsible for the trace gas measurements at the considered sites.

*Data availability:* While ICOS Near Real Time Observational Data (Level 1) and ICOS Final Fully Quality Controlled Observational Data (Level 2) can be obtained under licence CC-BY from the ICOS Carbon Portal (https://www.icos-cp.eu/), the 1-minute ICOS $CO_2$, $CH_4$ and CO data used for this exercise are not part of the ICOS official data releases and can be obtained by direct request to corresponding author.

*Competing interests*. The authors declare that they have no conflict of interest.

*Acknowledgements.* ICOS activities at CMN were supported by the Ministry of University and Research by the Joint Research Unit "ICOS Italia" throughout CNR-DSSTTA. Cosimo Fratticioli and Pamela Trisolino grants are funded by "Progetto nazionale Rafforzamento del Capitale Umano CIR01_00019 – PRO-ICOS-MED "Potenziamento della rete di
osservazione ICOS-Italia nel Mediterraneo – Rafforzamento del Capitale Umano" funded by the Ministry of University and Research. The observations at Jungfraujoch are part of ICOS Switzerland, which is supported by the Swiss National Science Foundation, in-house contributions, and the State Secretariat for Education, Research and Innovation.



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

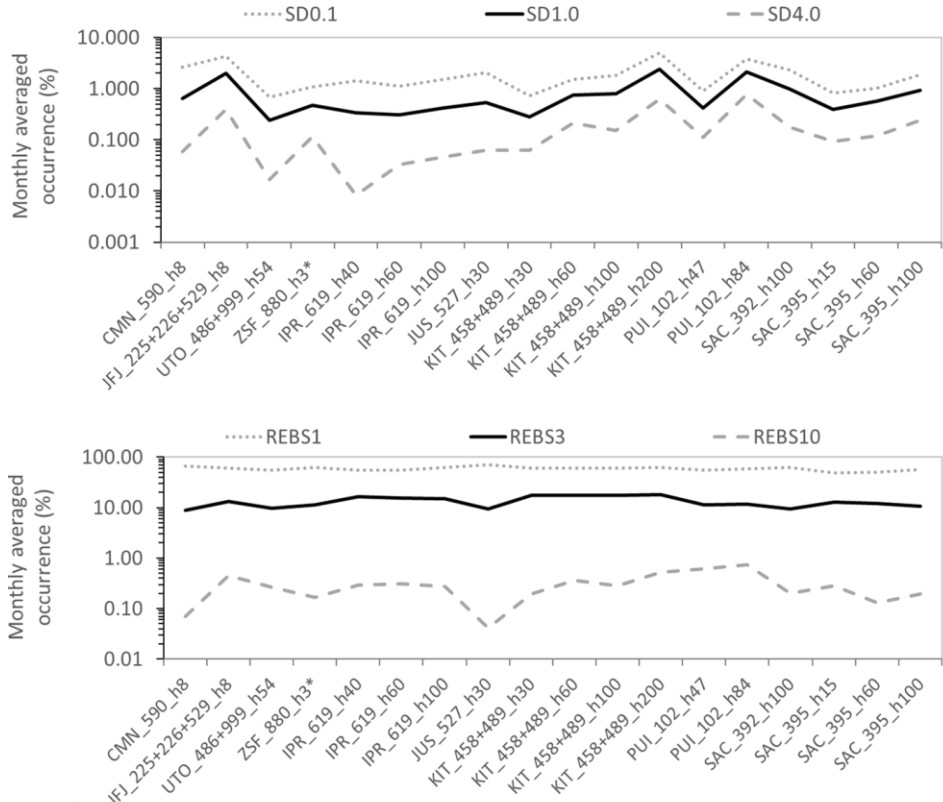


**Figure 1: Percentages of the 1-minute $CO_2$ data selected as spike by SD (upper plot) and REBS (bottom plot) over the period 2018 - 2020 (for ZSF_880_h3, only 2021 was considered), for the different algorithm settings. For SD, $\alpha$ was set to 0.1 (SD0.1), 1.0 (SD1.0) and 4.0 (SD4.0). For REBS, $\beta$ was set to 1 (REBS1), 3 (REBS3) and 10 (REBS10). The codes reported as labels for the x-axis indicate the combination among site, instrument(s) ID and sampling height (m a.g.l.).**






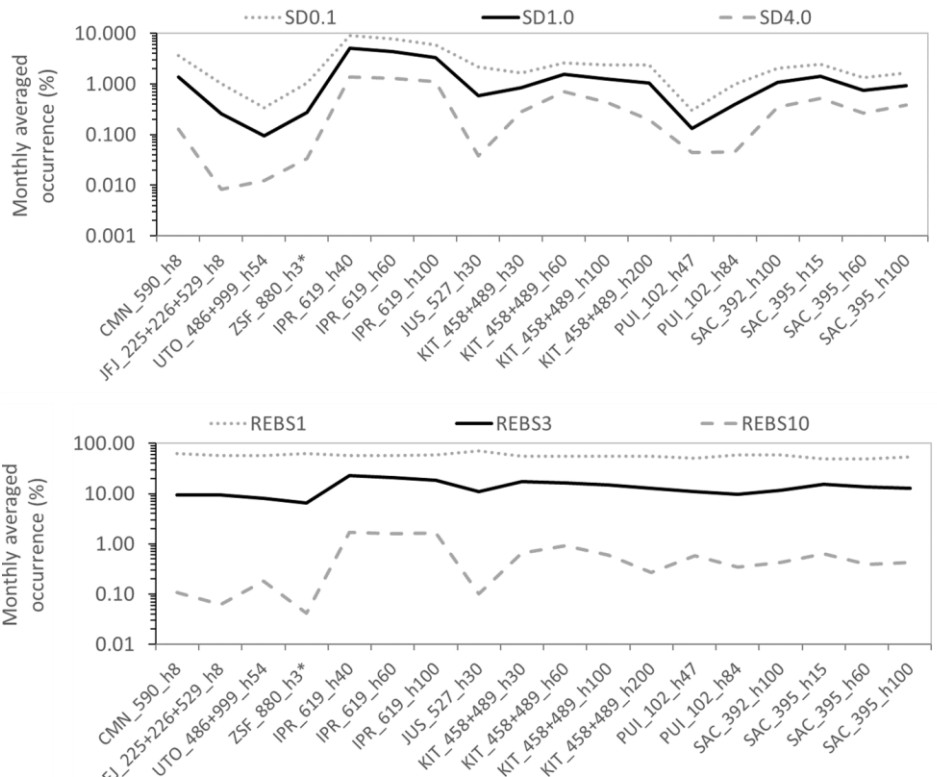

**Figure 2: As Figure 1 but for CH₄.**





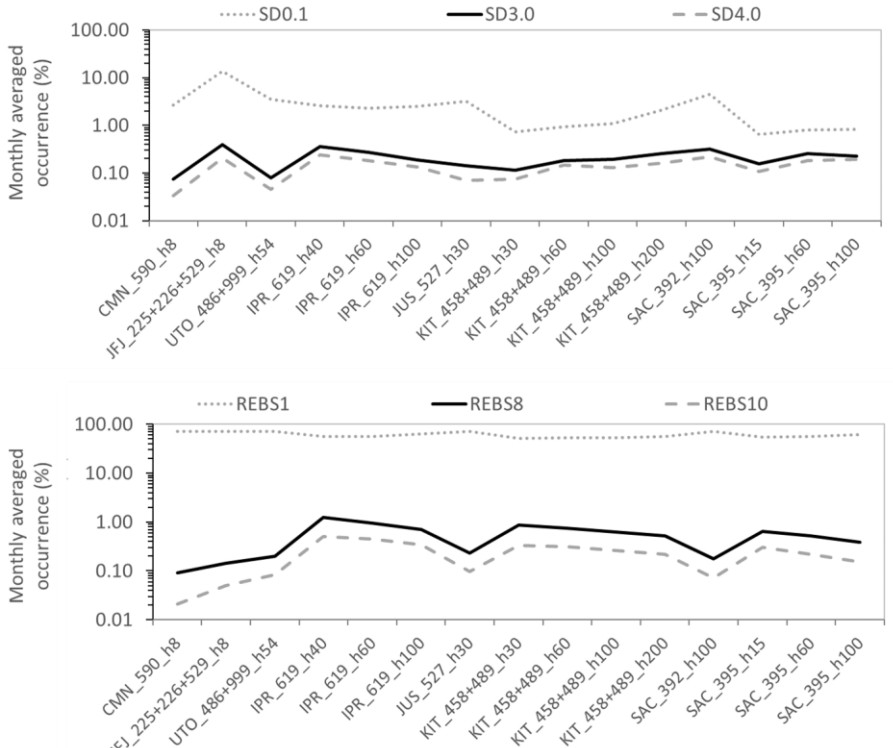

Figure 3: As Figure 1 but for CO.






**Figure 4: For the different sites (color code) the differences in the percentiles of hourly mean values between de-spiked and original dataset for the different species and methods are reported. The horizontal dotted lines represent the WMO network compatibility goal.**






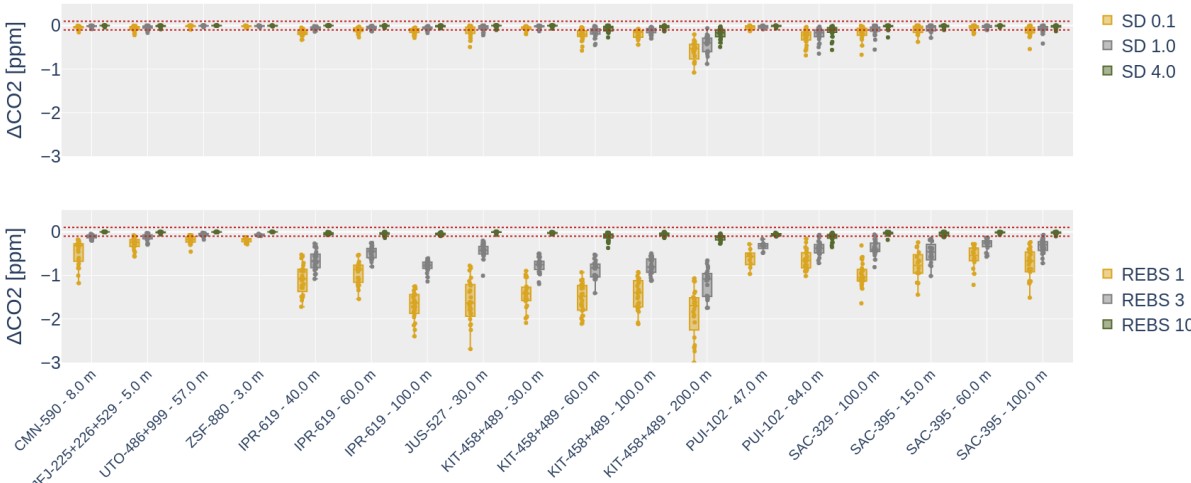

**Figure 5: Boxplot differences ($\Delta CO_2$) of the monthly mean $CO_2$ values between de-spiked and original dataset. The horizontal red lines represent the ±0.1 ppm WMO network compatibility goal. Each box represents the distribution of the differences for a combination of site and sampling height. The colour code indicates, for each algorithm, the adopted setting.**






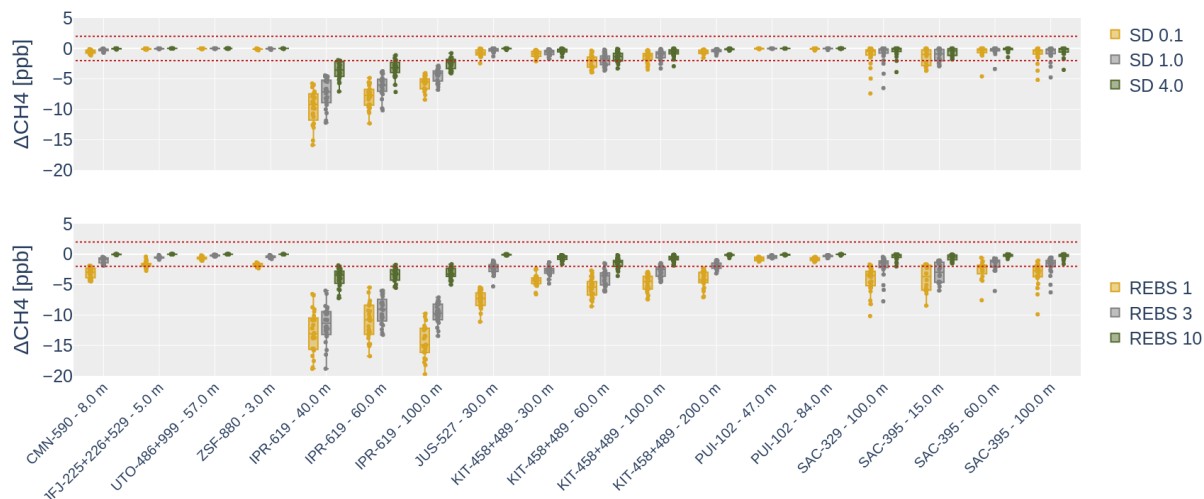

Figure 6: As Figure 4 but for CH₄. The horizontal red lines represent the ±2 ppb WMO network compatibility goal.





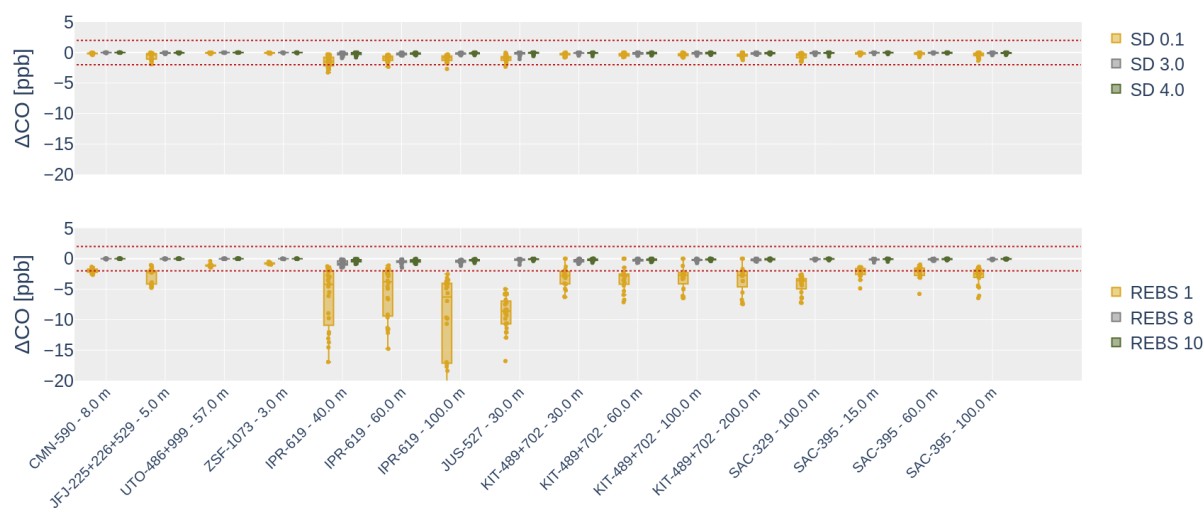


Figure 7: As Figure 4 but for CO. The horizontal red lines represent the ±2 ppb WMO network compatibility goal.







**Figure 8: Mean seasonal diurnal cycles of CO₂ at selected sites and sampling heights for different data selection: results for the**
**original data are shown ("non-spiked") together with those after de-spiking for SD with α = (0.1, 1, 4) and REBS with β = (1, 3,**
**10). The grey areas indicate the WMO network compatibility goal referred to the original dataset.**



**Figure 9: As for Figure 7 but for CH₄.**





**Figure 10: As for Figure 7 but for CO with de-spiking for SD with α = (0.1, 3, 4) and REBS with β = (1, 8, 10).**





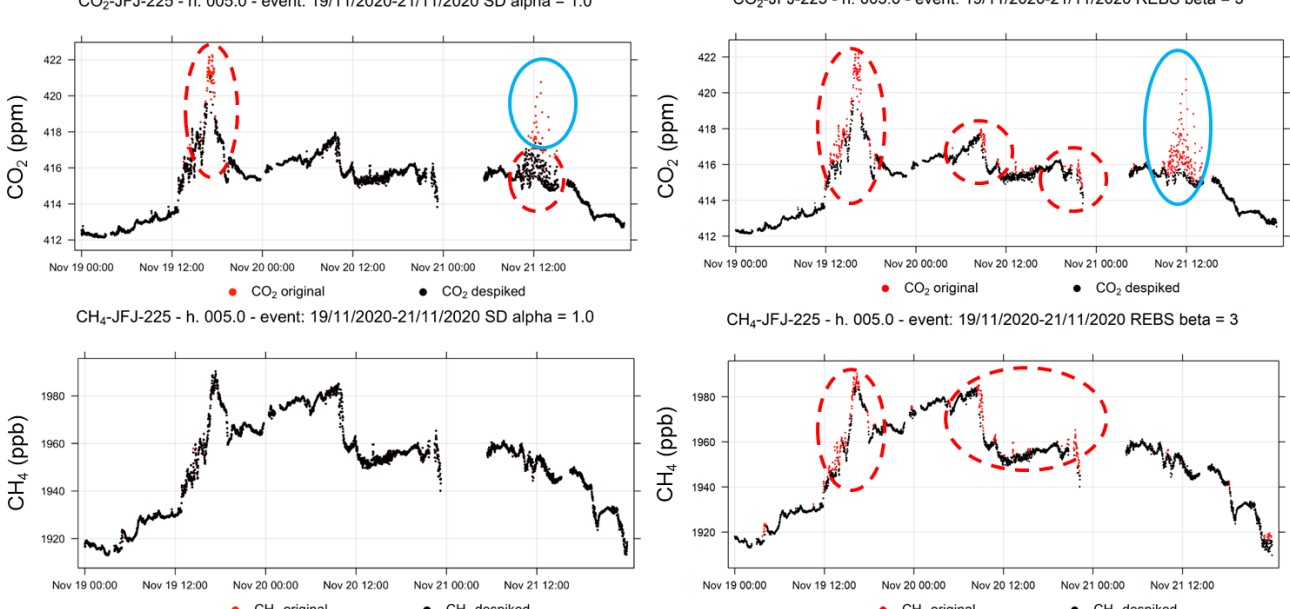

**Figure 11: CO₂ and CH₄ observations at JFJ (19-21 November 2020). No-spike data are reported by the black points ("despiked"); red points ("original") denote the data flagged as spikes using SD (left) and REBS (right). Continuous (dotted) circles represent the spike attribution manually confirmed (not confirmed) by site PI.**





**Figure 12: CO₂, CH₄ and CO observations at IPR (2 July 2019). No-spike data are reported by the black points ("despiked"); red points ("original") denote the data flagged as spikes using SD (left) and REBS (right). Continuous (dotted) circles represent the spike attribution manually confirmed (not confirmed) by site PI.**




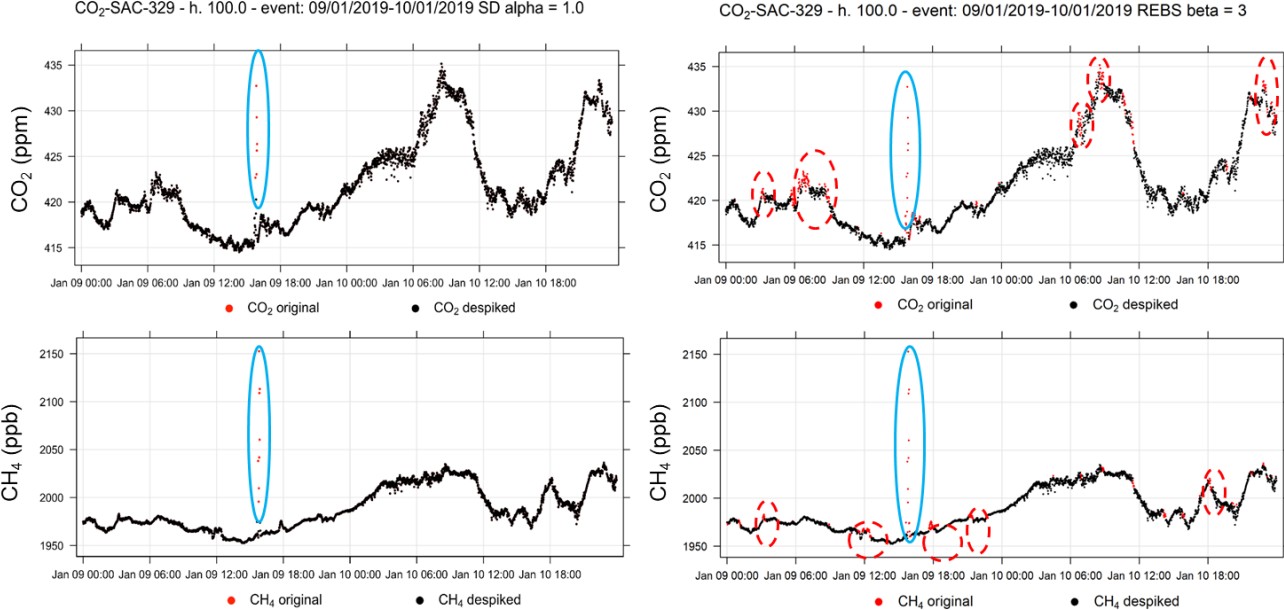

**Figure 13: CO₂ and CH₄ observations at SAC (9-10 January 2019). No-spike data are reported by the black points ("despiked"); red points ("original") denote the data flagged as spikes using SD (left) and REBS (right). Continuous (dotted) circles represent the spike attribution manually confirmed (not confirmed) by site PI.**





**Table 1: List of specific instruments working at the considered ICOS sites during the study period. Sites are listed in alphabetical order according with their classifications (remote; non-remote).**

| Classification | Site | Species | Instrument model #ICOS-ID; start date - end date |
|---|---|---|---|
| Remote | CMN | $CO_2$, $CH_4$, CO | Picarro G2401 #590; 1/1/2019 - 31/12/2020 |
| | JFJ | $CO_2$, $CH_4$ | Picarro G2401 #225; 1/1/2019 - 15/10/2019<br>Picarro G2401 #226; 15/10/2019 - 26/11/2020<br>Picarro G2401 #529; 23/12/2020 - 31/12/2020 |
| | | CO | Picarro G2401 #529; 23/12/2020 - 31/12/2020<br>Picarro G2401 #226; 15/10/2019 - 26/11/2020<br>Picarro G2401 #225; 1/1/2019 - 15/10/2019<br>LGR 913-0015 (EP) ; #412; 1/1/2019 - 7/1/2020 |
| | PDM | $CH_4$ | Picarro G2401 #222; 2014/11/1 - 2015/12/31 |
| | UTO | $CO_2$, $CH_4$, CO | Picarro G2401 #489; 1/1/2019 - 11/12/2020<br>Picarro G2401 #999; 11/12/2020 - 31/12/2020 |
| | ZSF | $CO_2$, $CH_4$ | Picarro G2301 #880; 13/1/2021 - 31/12/2021 |
| | | CO | LGR 913-0015 (EP) ; #1073 131/1/2021 - 18/12/2021 |
| Non-remote | IPR | $CO_2$, $CH_4$, CO | Picarro G2401 #619; 1/1/2019 - 31/12/2020 |
| | JUS | $CO_2$, $CH_4$, CO | Picarro G2401 #527; 1/1/2019 - 31/12/2020 |
| | KIT | $CO_2$, $CH_4$, CO | Picarro G2401 #489 31/1/2019 - 28/3/2019<br>Picarro G2301 #458 28/3/2019 - 31/12/2020 |
| | | CO | Picarro G2401 #489 31/1/2019 - 28/3/2019<br>LGR 913-0015 (EP) #702; 5/8/2019 - 31/12/2020 |
| | PUI | $CO_2$, $CH_4$ | Picarro G2301 #102; 1/1/2019 - 31/12/2020 |
| | SAC | $CO_2$, $CH_4$, CO | Picarro G2401 #329; 1/1/2019 - 31/12/2020<br>Picarro G2401 #395; 1/1/2019 - 31/12/2020 |
| | | CO | Picarro G2401 #395; 1/1/2019 - 31/12/2020<br>Picarro G5310 #781; 23/01/2019 - 31/12/2020 |





**Table 2. False alarm rate ("F"), hit rate ("H") and bias ("BIAS") from the comparison of automatic (standard settings) and manual spike identifications. The optimal values for α ("SD") and β ("REBS") leading to the maximum agreement with the manual flagging are also reported ("Best parameter"). The periods over which the test was carried out is reported in the second column ("Period") for each site, while the percentages of spike identified by the site PIs are reported in the third column (italic).**

| Station | Period | Species | Method | F | H | BIAS | Best parameter |
|---------|--------|---------|--------|------|------|------|----------------|
| IPR | April, 2019 July, 2020 | CH$_4$ *17.5%* | SD | *0.00* | 0.20 | 0.2 | 0.1 |
| | | | REBS | 0.10 | *0.58* | *1.1* | 5 |
| PDM | July-August 2015 | CH$_4$ *18.5%* | SD | *0.02* | *0.49* | 0.6 | 1.0 |
| | | | REBS | 0.13 | 0.43 | *1.0* | 3 |
| PUI | January, 2019 June, 2020 | CH$_4$ *14.6%* | SD | 0.00 | 0.02 | 0.02 | 0.1 |
| | | | REBS | 0.08 | 0.22 | 0.71 | 6 |
| | | CO$_2$ *14.6%* | SD | 0.00 | 0.14 | 0.15 | 0.1 |
| | | | REBS | 0.08 | 0.43 | 0.87 | 6 |
| UTO | April, 2019 June, 2020 | CH$_4$ *1.7%* | SD | 0.00 | 0.01 | 0.21 | 1.0 |
| | | | REBS | 0.09 | 0.17 | 5.36 | 3 |
| | | CO$_2$ *1.9%* | SD | 0.00 | 0.05 | 0.4 | 0.5 |
| | | | REBS | 0.10 | 0.22 | 5.5 | 7 |
| | | CO *1.8%* | SD | 0.00 | 0.01 | 0.1 | 1.0 |
| | | | REBS | 0.00 | 0.03 | 0.1 | 5 |





**Table 3. Same as Table 2 but with results from the high spikes analysis.**

| Station | Period | Species | Method | F | H | BIAS | Best parameter |
|---------|--------|---------|--------|------|------|------|----------------|
| IPR | April, 2019 July, 2020 | $CH_4$ *17.5%* | SD | 0.01 | 0.36 | 0.2 | 0.1 |
| | | | REBS | 0.12 | 0.89 | 2.2 | 6 |
| PDM | July-August 2015 | $CH_4$ *18.5%* | SD | 0.03 | 0.70 | 1.0 | 1.0 |
| | | | REBS | 0.12 | 0.48 | 1.0 | 3 |
| PUI | January, 2019 June, 2020 | $CH_4$ *14.6%* | SD | 0.00 | 0.16 | 0.2 | 0.1 |
| | | | REBS | 0.09 | 0.90 | 7.3 | 7 |
| | | $CO_2$ *14.6%* | SD | 0.01 | 0.42 | 0.6 | 0.5 |
| | | | REBS | 0.10 | 0.95 | 3.6 | 7 |
| UTO | April, 2019 June, 2020 | $CH_4$ *1.7%* | SD | 0.00 | 0.2 | 3.3 | 2.5 |
| | | | REBS | 0.09 | 0.93 | 83.9 | 10 |
| | | $CO_2$ *1.9%* | SD | 0.01 | 0.33 | 2.5 | 2.5 |
| | | | REBS | 0.10 | 0.84 | 36.5 | 10 |
| | | CO *1.8%* | SD | 0.00 | 0.09 | 0.3 | 2.0 |
| | | | REBS | 0.00 | 0.21 | 0.6 | 8 |
