# Peer review of "Identification of spikes in continuous ground-based in-situ time series of CO2, CH4 and CO: an extended experiment within the European ICOS-Atmosphere Network"

_Atmospheric Measurement Techniques, 2023_

## Author Comment (AC4)

[Figure]

**Figure 8: Mean seasonal diurnal cycles of CO₂ for different data selection and sampling heights at ZSF, IPR and KIT: results for the original data are shown ("non-spiked") together with those after de-spiking for SD with α = (0.1, 1, 4) and REBS with β = (1, 3, 10). The grey areas indicate the WMO network compatibility goal referred to the original dataset.**

[Figure]

**Figure 9: As for Figure 7 but for CH₄ at IPR, KIT, SAC 329 and SAC 395.**

---

## Author Comment (AC5)

**Table 4: List of specific method/setting currently adopted for each considered site. Sites are listed in alphabetical order according with their classifications (remote; not-remote).**

| Classification | Site | Species | Method, setting |
|---|---|---|---|
| Remote | CMN | $CO_2$, $CH_4$, CO | SD, $\alpha = 1$
 SD, $\alpha = 1$
 SD, $\alpha = 3$ |
| | JFJ | $CO_2$, $CH_4$
 CO | SD, $\alpha = 1$
 SD, $\alpha = 1$
 SD, $\alpha = 3$ |
| | UTO | $CO_2$
 $CH_4$
 CO | SD, $\alpha = 1$
 SD, $\alpha = 1$
 SD, $\alpha = 3$ |
| | ZSF | $CO_2$
 $CH_4$
 CO | SD, $\alpha = 1$
 SD, $\alpha = 1$
 SD, $\alpha = 3$ |
| Not-remote | IPR | $CO_2$
 $CH_4$,
 CO | REBS, $\beta = 3$
 REBS, $\beta = 3$
 REBS, $\beta = 8$ |
| | JUS | $CO_2$
 $CH_4$
 CO | SD, $\alpha = 1$
 SD, $\alpha = 1$
 SD, $\alpha = 3$ |
| | KIT | $CO_2$
 $CH_4$
 CO | REBS, $\beta = 3$
 REBS, $\beta = 3$
 REBS, $\beta = 8$ |
| | PUI | $CO_2$
 $CH_4$ | REBS, $\beta = 3$
 REBS, $\beta = 3$ |
| | SAC | $CO_2$
 $CH_4$
 CO | REBS, $\beta = 3$
 REBS, $\beta = 3$
 REBS, $\beta = 8$ |

---

## Author Comment (AC6)

**Table 1: List of specific instruments working at the considered test sites during the study period. Sites are listed in alphabetical order according with their classifications (remote; not-remote). ICOS-labelled sites are identified by an asterisk.**

| Classification | Site (acronym, country) | Sampling heights (above ground level) | Species | Instrument model #ICOS-ID; start date - end date |
|---|---|---|---|---|
| Remote | Monte Cimone * (CMN, Italy) | 8 m | $CO_2$, $CH_4$, CO | Picarro G2401 #590; 1/1/2019 - 31/12/2020 |
| | Jungfraujoch * (JFJ, Switzerland) | 5 m | $CO_2$, $CH_4$ | Picarro G2401 #225; 1/1/2019 - 15/10/2019 Picarro G2401 #226; 15/10/2019 - 26/11/2020 Picarro G2401 #529; 23/12/2020 - 31/12/2020 |
| | | | CO | Picarro G2401 #529; 23/12/2020 - 31/12/2020 Picarro G2401 #226; 15/10/2019 - 26/11/2020 Picarro G2401 #225; 1/1/2019 - 15/10/2019 LGR 913-0015 (EP) ; #412; 1/1/2019 - 7/1/2020 |
| | Pic du Midi (PDM, France) | 10 m | $CH_4$ | Picarro G2401 #222; 2014/11/1 - 2015/12/31 |
| | Utö - Baltic sea * (UTO, Finland) | 57 m | $CO_2$, $CH_4$, CO | Picarro G2401 #489; 1/1/2019 - 11/12/2020 Picarro G2401 #999; 11/12/2020 - 31/12/2020 |
| | Zugspitze * (ZSF, Germany) | 3 m | $CO_2$, $CH_4$ | Picarro G2301 #880; 13/1/2021 - 31/12/2021 |
| | | | CO | LGR 913-0015 (EP) ; #1073 131/1/2021 - 18/12/2021 |
| Not-remote | Ispra * (IPR, Italy) | 40 m, 60 m, 100 m | $CO_2$, $CH_4$, CO | Picarro G2401 #619; 1/1/2019 - 31/12/2020 |
| | Jussieu Paris (JUS, France) | 30 m | $CO_2$, $CH_4$, CO | Picarro G2401 #527; 1/1/2019 - 31/12/2020 |
| | Karlsruhe * (KIT, Germany) | 30 m, 60 m, 100 m, 200 m | $CO_2$, $CH_4$, CO | Picarro G2401 #489 31/1/2019 - 28/3/2019 Picarro G2301 #458 28/3/2019 - 31/12/2020 |
| | | | CO | Picarro G2401 #489 31/1/2019 - 28/3/2019 LGR 913-0015 (EP) #702; 5/8/2019 - 31/12/2020 |
| | Puijo * (PUI, Finland) | 47 m, 84 m | $CO_2$, $CH_4$ | Picarro G2301 #102; 1/1/2019 - 31/12/2020 |
| | Saclay * (SAC, France) | 15 m, 60 m, 100 m | $CO_2$, $CH_4$, CO | Picarro G2401 #329; 1/1/2019 - 31/12/2020 Picarro G2401 #395; 1/1/2019 - 31/12/2020 |
| | | | CO | Picarro G2401 #395; 1/1/2019 - 31/12/2020 Picarro G5310 #781; 23/01/2019 - 31/12/2020 |

---

## Author Response (AR1)

**Reviewer 1**

**General comments**:

In this paper the authors evaluated the effectiveness of two types of algorithms that can be used to automatically identify spikes in continuous long-term data sets. Their application was specifically focussed on GHG species and the observations within the ICOS network (which consists of various sites, each having unique site specific properties). The explanation of the two algorithms (one based on the standard deviation of the background - SD and the other a robust extraction of baseline signal - REBS) and their respective modifiable (tuneable) factors to the specific measuring stations, is of interest to the wider scientific community and certainly applicable to other non-ICOS measuring stations, who also produces continuous, long-term data records. The graphics in the paper are relevant and clearly shows the reader the observed differences between the applied filter(s) and original data set. This paper is well written and very logically structured. The conclusions that were arrived at are justified and well-articulated.

*AR: We thank the reviewer for her/his/their positive position about our manuscript. We also tank her/him/them for the useful suggestions that have been taken into consideration for modifying the paper (see our point-to-point replies in the following marked as "AR-Author Reply").*

**Specific comments**:

**P5 Line 150**: It would be beneficial if the authors can state the approximate numbers of tourists per days / season? This will provide the reader additional information on their possible impact at the sites (also for other sites mentioned where visitor platforms are in close proximity to instrument inlets …).
*AR: we have numbers for Jungfraujoch, Mt. Cimone and Puijo and we reported them in the revised manuscript. Section 2.1.2: "According with Affolter et al. (2021), about 1 million tourists is visiting the Jungfraujoch per year. Three floors of the Sphinx laboratory are only accessible for researchers with the inlet on the top of the laboratory and a public terrace below.". Section 2.1.4: ". During summer, tourists (roughly 500-600 persons/year for the July-August period) can access the terrace of the "O. Vittori" Observatory (approximately 5 m below the sampling inlet)." Section 2.1.9: "The Puijo tower is also accessible to tourists (roughly 100'000 persons/year, during the high season of June-August on average 800 persons/day) and has a viewing platform at ~15 m below the top inlet and ~22 m above the lower inlet."*

**P11 Line 332**: The authors mentioned "…again, significant impacts of the de-spiking were observed…" Was this "significance" statistically tested? It would add value to the

discussion if the authors could mention a confidence level / statistical evaluation of this significance.

*AR: we did not perform statistical significance test. As mentioned in the paper, we used the terms "impact" or "significant" when the differences between de-spiked and original dataset exceeded the WMO network compatibility goals for the averaged mole fractions (i.e. ± 0.1 ppm for $CO_2$, ± 2 ppb for $CH_4$ and CO; see World Meteorological Organization (2020)."*

*We removed the term "significant" from the analysis reported in Section 3.1 and 3.2 when describing the effect of the methodology in the number of detected spikes and to the changes in the percentiles of the mole fraction data distribution.*

A general comment - Have the authors considered making use of CO as a spike filter? It might be worthwhile for the authors to consider adding a second, independent spike filter parameter, such as CO to aid in the refining of the primary spike detection technique. This will certainly assist in instances where anthropogenic emissions are the source of these spikes. It might also only be quite site specific, and not applicable to a general solution...

*AR: we thank the reviewer for her/his/their interesting suggestion. As discussed in the Section 4, there is space for future refinements of the methodology and we already cited the possibility of combining spike detection on multiple species (CO was mentioned, instead). Since CO is a recommended parameter in ICOS, it can represent a suitable candidate for a first pilot test especially for $CO_2$. In the revised version of the manuscript, we added the possibility to test pre-filtering with tracers for anthropogenic emissions also taking advantage of chemical species observed in  the framework of other atmospheric research infrastructures: " A further action to be pursued is the exchange of experiences with other initiatives or measurement networks in the atmospheric composition landscape (e.g. the "Aerosol, Clouds and Trace Gases Research Infrastructure" - ACTRIS RI or the "Advanced Global Atmospheric Gases Experiment" - AGAGE) with the aim of considering different or novel (e.g. machine-learning based) spike detection methods or combing the information coming from different chemical species (e.g. synthetic compounds or $NO_x$) to improve the attribution of detected spikes."*

**Technical corrections**:

**P6 Line 164**: Sentence construction requires a re-write/rephrase... "...agricultural activities taking place during the livestock farming..." replace "taking place during the" with "from"
*AR: fixed*

**P10 Line 308**: Replace "Not" with "No"
*Fixed*

**P13 Line 403:** Rephrase "…by looking into CO also REBS…" with "…by looking at CO, REBS also …"
*Fixed*

**P13 Line 406**: Rephrase "…the most part of spike events…" with " …most of the spike events…"
*Fixed*

**P15 Line 474**: Replace "…not…" with "…no…"
*Fixed*

**Reviewer 2**

Cristofanelli et al. applied two automatic spike identification methods, SD and REBS, to continuous observations of $CO_2$, $CH_4$ and CO from ICOS-RI network. They conducted a comprehensive comparison of both methods for measurements across different time scales (hourly, monthly) at sites within various environments (remote, continental, urban). The manuscript is well-written and falls within the scope of AMT. Thus, I recommend its publication with a few minor revisions.

*AR: we thank the Reviewer for her/his/their positive position about our manuscript. We also tank her/him/them for the useful suggestions that have been taken into consideration for modifying the paper (see our point-to-point replies in the following marked by "AR" – "Author Reply").*

While I agree with the authors that the ability to detect spikes for each method is a function of the sites, events, and considered species, I strongly suggest that the authors compile a summary table at the end, outlining their recommendations for spike identification methods and the parameters utilized for each method at each site.

*AR: we have provided recommendations in the final section of the manuscript and added the suggested summary table ("Table 4 summarises the combination of methods/settings which have been operationally implemented at the analysed test sites by taking into consideration the results of this experiment and the need to standardise as much as possible the detection method among the sites") and explicit recommendations, e.g. "we recommend that REBS is implemented only for specific sites, mostly affected by more or less frequent very nearby local emissions (like IPR and PUI), where clear benefits in using REBS were demonstrated.", "…, it is recommendable to perform sensitivity experiments to evaluate and document the performance of the implemented detection method, including its setting."*

1. The figures are dense, especially Figures 8-10. Please simplify it and consider moving some subplots into supplements.

*AR: according with the reviewer's comment, we moved Figure 8-10 in the SM. We only kept in the main manuscript a sub-set of representative plots to support the description of the results.*

2. Line 70-71, it is recommended to add the definition of "regional" signal and to ensure a distinct differentiation between "local" and "regional" signals.

*AR: we thank the reviewer for arising this point. In our case, with the term "regional" we refer to emission occurring in a range of ~100-500 km from the measurement sites, according with the spatial scale definition provided in Oney et al. (2015) and fitting with the general*

*framework provided by Storm et al. (2023) for the definition of an appropriate atmospheric network for inclusion in "regional" inversion studies. Moreover, we considered local emissions, those occurring in a range of ~ 10 km from the measurement site (El Yazidi et al., 2018). We added this information in the manuscript (Section 1: "Here, "local" refers to emissions occurring in a range of a few km (i.e., ~ 10 km) from the site which cause positive short-term spikes with maximum duration from minutes to less than a few hours. These signals are not suitable for investigating regional-scale (~ 100–500 km) fluxes within the site surface sensitivity area (Oney et al., 2015; Storm et al., 2023) because they are superimposed to the GHG variability resulting from the atmospheric background and the regional signal.").*

*However, it's challenging to define a sharp single differentiation. As an instance, taking in mind the use of the ICOS atmospheric observations for "regional" inversion studies, the differentiation between "local" and "regional" can be related to the model capacities in diagnosing the transport of emissions to the measurement sites. Thus, this definition can thus dependent on the modelling system adopted. Moreover, due to the different local meteorology, topography and type of emission sources, the observations at each measurement site can respond in different ways to a near release of emissions in atmosphere, making the definition of a common distinct differentiation between "local" and "regional" signals very complex.*

*El Yazidi, A., Ramonet, M., Ciais, P., Broquet, G., Pison, I., Abbaris, A., Brunner, D., Conil, S., Delmotte, M., Gheusi, F., Guerin, F., Hazan, L., Kachroudi, N., Kouvarakis, G., Mihalopoulos, N., Rivier, L., and Serça, D.: Identification of spikes associated with local sources in continuous time series of atmospheric CO, CO2 and CH4, Atmos. Meas. Tech., 11, 1599–1614, https://doi.org/10.5194/amt-11-1599-2018, 2018.*

*Oney, B., Henne, S., Gruber, N., Leuenberger, M., Bamberger, I., Eugster, W., and Brunner, D.: The CarboCount CH sites: characterization of a dense greenhouse gas observation network, Atmos. Chem. Phys., 15, 11147–11164, https://doi.org/10.5194/acp-15-11147-2015, 2015.*

*Storm, I., Karstens, U., D'Onofrio, C., Vermeulen, A., and Peters, W.: A view of the European carbon flux landscape through the lens of the ICOS atmospheric observation network, Atmos. Chem. Phys., 23, 4993–5008, https://doi.org/10.5194/acp-23-4993-2023, 2023.*

Line 104: Change "to of access" to "to access"

*OK*

**Reviewer 3**

The paper presents a systematic analysis of spike detection algorithms applied to ICOS atmospheric data for CH4, CO2, and CO, for a variety of sites. In general the paper is well written, and I recommend publication after the following concerns have been addressed.

*AR: we thank the reviewer for her/his/their positive evaluation of our manuscript. We also tank her/him/them for the useful suggestions that have been taken into consideration for improving the paper (see our point-to-point replies in the following marked by "AR" – "Author Reply").*

**General Comments:**

As the authors correctly point out in the introduction, it is "very local emissions" that are of concern when using the data in inverse atmospheric transport models. I miss a discussion on this main use of ICOS atmosphere data in the discussion section. The basic question that needs discussion is what we expect models to represent. For example, the "very local sources of CH4 due to the systematic venting of cattle farms located in the proximity of the site" (Lines 374-375) for IPR could be included in a atmospheric transport model, of the resolution is sufficiently high.

*AR: we thank the reviewer for this question. The Integrated Carbon Observation System (ICOS) was designed as the European in situ observation and information system to support science and society in their efforts to mitigate climate change. The main use of ICOS data (including atmospheric data) have been already described by Heiskanen et al. (2021). The implementation strategy for any automatic spike detection methodology will be to flag the data as "spike" without removing them from the data collection. This leaves to the users to decide if using the information related to the original dataset or the de-spiked dataset. Referring to the specific case pointed out by the reviewer, this implies that data affected by local emissions are still available in the data collection and they can be used in the case an atmospheric transport model with sufficiently high spatial resolution is able to diagnose local emission occurring at very small spatial scale. We added a specific comment on the Section 4: "The strategy for the implementation of these automatic spike detection methods is to flag the 1-minute data without removing them from the data collection: this give to the data users the opportunity to decide if considering the original dataset or the de-spiked dataset depending by the specific data usage purposes."*

Furthermore, there was a discussion of using buffer volumes to time-integrate samples at ICOS sites with multiple vertical levels, such that meaningfull instantaneous gradient information can be obtained. This is also mentioned in the cited ICOS RI 2020 (Atmosphere Station Specifications V2.0). It should be at least mentioned, how many

ICOS sites are actually using buffers. Unfortunately it is unclear if any sites are using this as the meta data available through ICOS-CP don't seem to include any information on the use of a Buffer volume (although recommended in ICOS RI 2020). If there are ICOS sites where buffer volumes are deployed, it should be discussed that for those a different strategy needs to be deployed for filtering (e.g. de-convolution as demonstrated by Winderlich et al. (2010), followed by spike detection).

Reference: Winderlich, J., Chen, H., Gerbig, C., Seifert, T., Kolle, O., Lavrič, J. V., Kaiser, C., Höfer, A., and Heimann, M.: Continuous low-maintenance CO2/CH4/CO measurements at the Zotino Tall Tower Observatory (ZOTTO) in Central Siberia, Atmos. Meas. Tech., 3, 1113–1128, https://doi.org/10.5194/amt-3-1113-2010, 2010.

*AR: within the ICOS atmospheric network, the only stations currently used buffer volume are Svartberget (SVB), Norunda (NOR), Hyltemossa (HTM), Hyytiälä (SMR) and Cabauw (CBW). We appreciate the reviewer suggestion to clearly make the use of buffer volume visible in the station metadata. This is an information that should be easily available to external users. There was discussion in the ICOS Atmosphere Monitoring Station Assembly about the use of buffer volumes and the possibility to apply deconvolution. At the moment we still not have defined guidelines for this but in the Atmospheric Specification document (https://doi.org/10.18160/GK28-2188,) it is stated that "sites which are facing regular and significant local contamination should not use buffer volume as the spike detection (only suitable without buffer) impact must be higher than the representativeness error related to the data discretization (multi sampling height sampling without buffer)."*

*We also appreciated the suggested strategy for a possible application of de-spiking to the stations equipped with buffer volume. We added a specific comment in the revised version of Section 4: "Moreover, a specific strategy should be developed for the few stations within the ICOS network using buffer volumes (i.e. Svartberget, Norunda, Hyltemossa, Hyytiälä and Cabauw). One possibility is to implement deconvolution (see Winderlich et al., 2010), followed by the application of the spike detection."*

**Specific comments**

Ln 47: "networks with surface footprints representative-enough of the tagged spatial regions" may be replace with "networks whose surface footprints are representative enough of the tagged spatial regions "
*OK*

Ln 49: "the measurement sites must carry out accurate measurements" -> "accurate measurements are required at the measurement sites"
*OK*

Ln 59-62: Is the only objective of the near-real time delivery the application of QA/QC checks? I would hope that the driving idea behind is utilization of the data in NRT.
*AR: also the creation of a "growing" NRT dataset is included, of course. Now this info is reported in the revised version of the manuscript together with citation to the NRT data collection (see Section 1): "Also the data treatment follows standardized and centralized procedures: all raw data recorded by the measurement sites are delivered in near-real time (i.e. with a 24-hour delay) to the ICOS Atmospheric Thematic Center (ATC) for the application of automatic quality checks, the averaging to 1-minute and 1-hour mean values (Hazan et al., 2016) and the release of the ICOS Near Real-Time data collection (ICOS RI, 2018)."*

Ln 79: "basing" -> „based"
OK

Ln 104: „were used to of assess" drop the "of"
 OK

Ln 107: before using site abbreviations (here "PUI") I suggest using a reference to Table 1, to which I recommend adding the site names and countries as additional columns.

*AR: thanks for the suggestion. This info has been added to Table 1 of the revised manuscript. In the revised manuscript, we made reference to Table 1 before introducing the single measurement sites.*

Ln 218: "but not at PUI" -> „except for PUI, where inly CO2 and CH4 are measured"
OK

Ln 231: What is meant by "combination", one with "and" or with "or"? Or, in other words, are all spikes considered as spikes, or only those that are detected in both directions?
*AR: it is "or", actually. This is clarified in the revised manuscript: "The method was applied on the 1-minute data with two modes, forwards and backwards: all the detected spikes are kept."*

Ln 233: What happens if there is an interruption of data (e.g. the sampling switches to a different level), is "n" then still only counting data points, or is it counting time in minutes?
*AR: the current algorithm set-up takes into account data points.*

Table 1: In addition to site names and countries, also the sampling levels should be included as additional columns.
*AR: this information is now added in the revised Table 1.*

Ln 274: "Continental sites" – does this mean sites having a certain distance to the coast? Or non-mountain and non-island sites? Table 1 lists as site classification only "Remote" and "Non-remote", may be one can add also "continental" vs. others (coastal, island, mountain). Now going through the paper again, I see "continental" is described as environmental conditions, with others being "remote" and "urban". In that context may be this means "continental background"? There is a need for a clear characterization and nomenclature.

*AR: for continental sites, we refer to the ICOS classification ([https://doi.org/10.18160/GK28-2188](https://doi.org/10.18160/GK28-2188)) i.e., stations targeting predominantly continental air-masses. By "remote", we indicated sites less directly and less frequently exposed to air-masses carrying strong anthropogenic emissions (like mountain or coastal sites). We think that for the analysis presented in this manuscript, this categorization is reasonable. We better clarified the meaning of the adopted nomenclature in the revised manuscript (see Section 2.1): "With the term "remote", here we considered sites which are less directly and less frequently exposed to strong anthropogenic emissions. "Continental" indicates stations targeting predominantly continental air-masses, while "urban" indicates stations located in metropolitan districts. In this paper, we adopted this site classification because we expected different occurrence of spikes as a function of the more or less direct exposition to anthropogenic emissions."*

Fig. 4 caption: "differences in the percentiles of hourly mean values between de-spiked and original dataset" – I would call this "the percentiles of hourly mean value differences between de-spiked and original dataset". "differences in the percentiles" would not have units of ppb or ppm.

*AR: figure 4 reports the differences in the percentiles not the percentiles of differences, actually. This metric has been used to guarantee the consistency of the comparison between the original and the de-spiked datasets. The differences in the percentiles of the two datasets is expressed as ppm or ppb, actually: the underlying idea is to report the shift of the data population after the application of the de-spiking methods.*

Fig. 4: I suggest for easier comparisons to use the same y-axis range for the REBS and SD results.
*AR: Figure 4 was revised as suggested by the reviewer.*

Ln 388: "which method was in better agreement" please add with what this agreement is better. Is it the expectation by the expert as in indicated in the following sentence?
*AR: yes. This is now rephrased in the revised manuscript for better understanding: "When possible, we also provided an evaluation about which method was in better agreement with the subjective judgement of the stations PIs for these specific case studies. To this aim, we varied the standard configurations (α and β values) and provided the optimal method configuration based on the subjective PI inspection of the de-spiking method results for each case studies."*

Ln 394: what is meant by " "standard" settings at JFJ"?
*AR: "standard" settings for SD and REBS have been defined in Section 3.1. We added a reference to this Section for increasing the readability ("(see Section 3.1 for the definition of "standard" settings for SD and REBS)"*

Ln 454: "this exercise would allow" may be replace with "this exercise allows"?
*OK, changed.*

Ln 468-469: note that in Ln 451-452 it is stated that "BIAS" should be as close as possible to 1. With the exception of site UTO REBS shows thus better (not just "much higher") BIAS than SD.
*AR: we rephrased: "REBS had better BIAS than SD."*

Ln 475: may be replace "In respect to the all-spike analysis, both SD and REBS were more effective in catching events" with "Compared with the all-spike analysis, both SD and REBS were more effective in catching high-spike events" (if I understood this correctly). Also please refer to Table 3.
*AR: we rephrased and we added a reference to Table 3*

Ln 528: "running REBS on standard deviation records" this is not clear to me
*AR: instead of running REBS on the 1-minute average values of mole fraction, it has been run on the 1-minute standard deviation to tentatively point out periods characterized by high short-term variability. We rephrased as following: "suggesting that running REBS on the time*

*series of 1-minute standard deviations was not a suitable strategy to automatically detect spikes."*